# PTEN self-regulates through USP11 via the PI3K-FOXO pathway to stabilize tumor suppression

Mi Kyung Park[1], Yixin Yao[1], Weiya Xia[1], Stephanie Rebecca Setijono[2], Jae Hwan Kim[1,3], Isabelle K. Vila[1], Hui-Hsuan Chiu[1], Yun Wu[4], Enrique González Billalabeitia[5], Min Gyu Lee[1], Robert G. Kalb[6], Mien-Chie Hung[1,7,8], Pier Paolo Pandolfi [9], Su Jung Song[2] & Min Sup Song[1,7]

PTEN is a lipid phosphatase that antagonizes the PI3K/AKT pathway and is recognized as a major dose-dependent tumor suppressor. The cellular mechanisms that control PTEN levels therefore offer potential routes to therapy, but these are as yet poorly defined. Here we demonstrate that PTEN plays an unexpected role in regulating its own stability through the transcriptional upregulation of the deubiquitinase USP11 by the PI3K/FOXO pathway, and further show that this feedforward mechanism is implicated in its tumor-suppressive role, as mice lacking *Usp11* display increased susceptibility to PTEN-dependent tumor initiation, growth and metastasis. Notably, *USP11* is downregulated in cancer patients, and correlates with PTEN expression and FOXO nuclear localization. Our findings therefore demonstrate that PTEN-PI3K-FOXO-USP11 constitute the regulatory feedforward loop that improves the stability and tumor suppressive activity of PTEN.

[1] Department of Molecular and Cellular Oncology, The University of Texas MD Anderson Cancer Center, Houston, TX 77030, USA. [2] Soonchunhyang Institute of Medi-bio Science, Soonchunhyang University, Cheonan-si, Chungcheongnam-do 31151, Republic of Korea. [3] Department of Biomedical Sciences, Seoul National University College of Medicine, Houston, Seoul 03080, Republic of Korea. [4] Department of Pathology, The University of Texas MD Anderson Cancer Center, Houston, TX 77030, USA. [5] Department of Clinical Oncology, Hospital Universitario Morales Meseguer-IMIB, Universidad Católica San Antonio de Murcia-UCAM, Murcia 30007, Spain. [6] Division of Neurology, Department of Pediatrics, Research Institute, Children's Hospital of Philadelphia, University of Pennsylvania, Philadelphia, PA 19104, USA. [7] Cancer Biology Program, The University of Texas Graduate School of Biomedical Sciences, The University of Texas MD Anderson Cancer Center, Houston, TX 77030, USA. [8] Center for Molecular Medicine and Graduate Institute of Cancer Biology, China Medical University, Taichung 404, Taiwan. [9] Cancer Research Institute, Beth Israel Deaconess Cancer Center, Department of Medicine and Pathology, Beth Israel Deaconess Medical Center, Harvard Medical School, Boston, MA 02215, USA. Correspondence and requests for materials should be addressed to S.J.S. (email: ssong1@sch.ac.kr) or to M.S.S. (email: msong1@mdanderson.org)

PTEN (phosphatase and tensin homolog) negatively regulates the highly oncogenic PI3K/AKT pathway through dephosphorylation of phosphoinositide-3,4,5-triphosphate (PIP3)[1,2]. Loss of PTEN function leads to a potent upregulation of the PI3K/AKT pathway, which stimulates cell growth, proliferation, migration, survival, and metabolism by phosphorylating the downstream signaling proteins such as FOXO transcription factors[3].

Many modeling efforts in *Pten* knockout mice have demonstrated that PTEN functions in a haplo-insufficient manner. Notably, the analysis of a series of hypomorphic *Pten* mouse models has revealed that even subtle reductions in PTEN dosage lead to an increased cancer susceptibility and higher rates of tumor progression[4,5]. These observations have inspired a new 'continuum model for tumor suppression' that integrates and updates Knudson's two-hit theory[6,7]. Furthermore, recent studies have shown that an increased PTEN dosage unexpectedly results in viable mice displaying a tumor-resistant, anti-Warburg metabolic state[8,9], implying that PTEN elevation may potentially represent a generally therapeutic approach in cancer. Intriguingly, whereas less than 5% of the sporadic breast tumors harbor *PTEN* mutations[10], a loss of PTEN protein immunoreactivity is found in nearly 40%[11]. Moreover, only 25% of cancer patients portray a correlation between the loss of PTEN protein and its mRNA level[12]. These data suggest that post-translational regulation of PTEN may contribute substantially to the development of human cancer.

Researchers have begun to identify the players in these post-translation processes. Recent studies have shown that the ubiquitin-proteasome system (UPS) is essential for the down-regulation of PTEN, and it has been proposed that the E3 ubiquitin ligases NEDD4-1, XIAP, WWP2, and CHIP mediate PTEN poly-ubiquitination and degradation[13–16]. In contrast, HAUSP, ataxin-3, USP13, and OTUD3 have all been identified recently as PTEN deubiquitinases (DUBs): HAUSP specifically removes the mono-ubiquitination of PTEN for its nuclear export[17], ataxin-3 regulates PTEN at the transcriptional level[18], and USP13 and OTUD3, which predominantly reside in the cytoplasm, affect cytosolic PTEN stability in a breast cancer-specific context[19,20]. While it is not surprising that such an important tumor suppressor is controlled by multiple DUBs, the physiological context of PTEN stability is yet to be addressed.

Ubiquitin-specific protease 11 (*USP11*, also known as *UHX1*) was originally identified as one of the X-linked retinal disorder genes at Xp11.23[21], although it is worth noting that a common deletion within the *USP11* interval had been observed earlier in ovarian cancer[22]. X-linked tumor suppressor genes are of particular interest because loss-of-heterozygosity (LOH) or mutation of a single allele can in effect functionally silence a gene[23]. As a deubiquitinase, USP11 is likely to have multiple protein substrates, such as p53, PML, and IκBα[24–26]. However, there is insufficient direct genetic evidence to define its precise role with the specificity required to target proteins of USP11 involved in tumorigenesis.

In this study, we report the identification of a PTEN feedforward mechanism and define both its critical role in tumorigenesis and its clinical relevance to patients.

## Results
**USP11 antagonizes PI3K activity by upregulating PTEN.** In order to identify DUBs that regulate the PI3K/AKT pathway, we first screened a synthetic siRNA library, targeting mouse DUBs in mouse embryonic fibroblasts (MEFs), and examined the rates of AKT phosphorylation (pS473 and pT308) using AlphaScreen assays (Supplementary Fig. 1a). We subsequently assessed the cellular levels of both PIP3, which is mainly found on the leading edges of filopodia and lamellipodia[27], and PTEN protein in cells expressing potential positive DUB shRNA vectors (Supplementary Fig. 1b, c). After target deconvolution of the observed hits, we identified USP11 as a potent inhibitor of the PI3K/AKT pathway on the basis of PTEN protein accumulation (Fig. 1a, b).

We then determined the binding between endogenous USP11 and PTEN by performing reciprocal immunoprecipitations (Fig. 1c) and confirmed their physical interaction by utilizing glutathione S-transferase (GST) pull-down assays (Supplementary Fig. 2a). Notably, wild-type (WT) USP11 deubiquitinated PTEN in vivo and in vitro, whereas the catalytically inactive USP11 C318S (CS) mutant, which could still bind to PTEN, exerted a dramatically diminished ability to deubiquitinate PTEN, and even exhibited a more heavy ubiquitinating effect (Fig. 1d and Supplementary Fig. 2b–e). Overexpression of WT, but not C318S, USP11 also increased PTEN levels (Fig. 1e). To further determine whether USP11 can regulate the stability of PTEN protein, we next examined endogenous PTEN protein levels in the presence of cycloheximide (CHX), an inhibitor of protein synthesis. Importantly, the stability of endogenous PTEN protein in NB4 human APL cells ($t_{1/2} \approx 12$ h) was significantly reduced upon the downregulation of USP11 ($t_{1/2} < 4$ h) (Fig. 1f).

Other DUBs, including USP13 and OTUD3, were recently found to stabilize PTEN protein through deubiquitination[19,20]. We then compared the PTEN-stabilizing ability of USP11, USP13, and OTUD3 by performing side-by-side assays. We first examined the interactions among endogenous PTEN, USP11, USP13, and OTUD3. PTEN and DUBs were readily co-immunoprecipitated with each other; however, USP11, USP13, and OTUD3 were not associated with each other (Supplementary Fig. 3a). Overexpression and knockdown experiments both revealed that USP11, USP13, and OTUD3 exhibit a synergistic effect on PTEN expression (Supplementary Fig. 3b, c). The ectopic expression of USP11 in *PTEN*-proficient human prostate cancer cell line DU145 depleted of USP13 or OTUD3 still upregulated PTEN (Supplementary Fig. 3d, e), suggesting that USP11 could be a more potent regulator of PTEN than other DUBs in our assays. Notably, the destabilization of PTEN in the nucleus upon the depletion of USP11 was more significant than that upon the loss of USP13 or OTUD3 (Supplementary Fig. 3f–i), suggesting that USP11, but not USP13 or OTUD3, regulates PTEN expression in the nucleus. Furthermore, our analysis of TCGA datasets from cBioPortal (www.cbioportal.org) and Kmplot (http://kmplot.com/analysis/)[28–31] indicates that the clinical significance of *USP11* is more significant than that of *USP13* or *OTUD3* (Supplementary Fig. 4).

We have previously demonstrated that HAUSP specifically removes the mono-ubiquitination of PTEN for its nuclear export[17]. We, therefore, tested whether USP11 could remove the mono-ubiquitination of PTEN in the nucleus. While HAUSP removed the mono-ubiquitination of PTEN, USP11 did not abolish the mono-ubiquitination of PTEN (Supplementary Fig. 3j), suggesting that USP11 represents a unique DUB that de-polyubiquitinates and stabilizes PTEN in the nucleus.

To further test the propensity of USP11 to antagonize PI3K activity via PTEN, we depleted Usp11 by RNA interference in *Pten*$^{+/+}$ and *Pten*$^{-/-}$ MEFs. Insulin stimulation led to acute AKT phosphorylation, as well as an increase of PIP3 in Usp11-silenced cells to levels higher than those observed in *Pten*$^{+/+}$ control cells, but not *Pten*$^{-/-}$ MEFs, which implies that USP11 has a PTEN-dependent PI3K inhibitory function (Fig. 1g, h). Similar results were observed upon the stimulation of serum, IGF-1, or EGF. Taken together, these results suggest that USP11 reverses poly-ubiquitination of PTEN, stabilizes PTEN protein, and thereby suppresses the activation of the PI3K/AKT pathway.

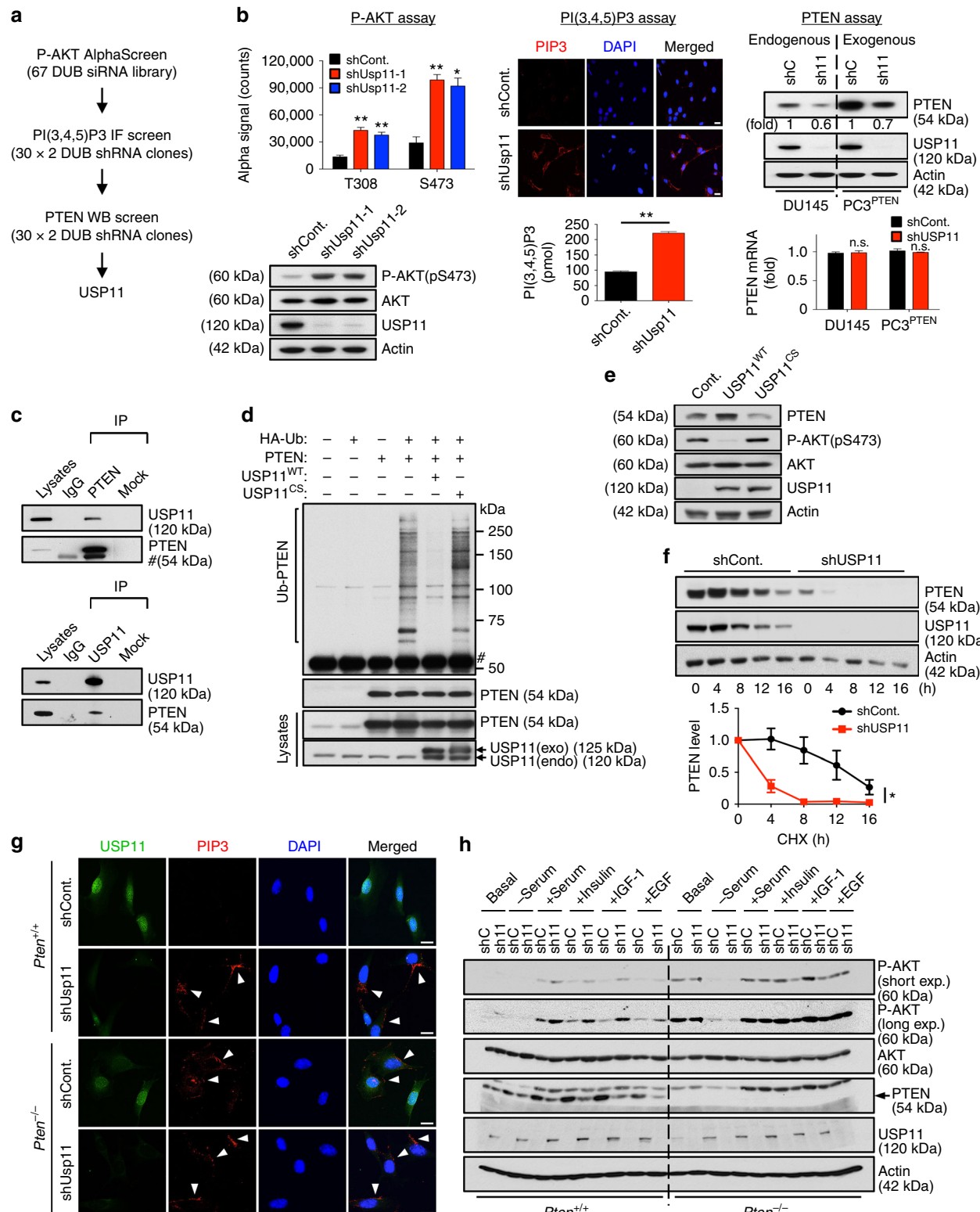

**Usp11 loss induces cell growth, motility, and metabolism.** To examine the biological functions of USP11 in vivo, we derived mice from a cryopreserved knockout mouse sperm (TF2623; Taconic Biosciences) and an oocyte donor. The gene-targeting cassette contained *lacZ*, the neomycin resistance gene (*neoR*), and had been integrated into exons 3–12 of *Usp11* to generate a dysfunctional allele that we named *Usp11⁻* (Supplementary Fig. 5a). As *Usp11* is an X-linked gene, hemizygous male ($Usp11^{-/Y}$) embryos were sufficient to create a *Usp11* loss-of-function mutation (Supplementary Fig. 5a, b). The levels of endogenous PTEN protein (but not mRNA) were reduced, and the phosphorylation rates of AKT and its downstream targets, such as FOXO1 and AS160 (or TBC1D4), were enhanced in primary MEFs isolated from $Usp11^{-/Y}$ embryos compared to WT cells (Supplementary Fig. 5b, c). Likewise, PTEN immunoprecipitates showed a heavier poly-ubiquitination pattern in

**Fig. 1** USP11 reduces PIP3 levels by deubiquitinating and stabilizing PTEN. **a** The screen scheme for the DUB library. **b** Validation of an RNAi screen in (**a**). MEFs expressing two independent Usp11 shRNAs were subjected to AlphaScreen assays (top, left) and immunoblotting (IB) (bottom, left). The levels of PIP3 in MEFs expressing Usp11 shRNA were evaluated using an IF (top, middle) and PIP3 Mass ELISA assays (bottom, middle). Lysates and total RNAs from *PTEN*-proficient DU145 cells and *PTEN*-deficient PC3 cells complemented with PTEN expressing USP11 shRNAs were subjected to IB (top, right) and RT-qPCR (bottom, right). Scale bars, 10μm. $n = 3$. Error bars represent ± SEM. $p$ Value was determined by Student's $t$ test (n.s., non-significant; *$p < 0.05$; **$p < 0.01$). **c** Lysates from DU145 cells were immunoprecipitated (IP) without (Mock) or with anti-PTEN (top) or anti-USP11 (bottom) antibody and subjected to IB. # indicates heavy chain of IgG. **d** Lysates from 293T cells transfected as indicated and treated with 10 μM MG132 for 4 h were IP with anti-Myc–PTEN, and the resulting immunoprecipitates were subjected to IB. # indicates the heavy chain of IgG. HA-Ub, HA-tagged ubiquitin. **e** Lysates from DU145 cells overexpressing wild-type (WT) or catalytically inactive C318S (CS) mutant of USP11 were subjected to IB. **f** Lysates from NB4 cells expressing USP11 shRNA and treated with cycloheximide (CHX, 100 μg ml$^{-1}$) for the indicated times were subjected to IB (top). PTEN protein levels were quantified by normalizing to the intensity of the actin band (bottom). $n = 3$. Error bars represent ± SEM. $p$ Value was determined by ANOVA (*$p < 0.05$). **g** IF analysis of PIP3 in *Pten*$^{+/+}$ and *Pten*$^{-/-}$ MEFs expressing Usp11 shRNAs starved for 8 h and stimulated with 100 nM insulin for 5 min. Arrowheads indicate the accumulation of PIP3 at the leading edges of membrane projections. Scale bars, 10μm. **h** Lysates from *Pten*$^{+/+}$ and *Pten*$^{-/-}$ MEFs expressing Usp11 shRNAs, starved for 8 h and stimulated with serum for 10 min or 100 nM insulin, 100 ng ml$^{-1}$ IGF-1, or 25 ng ml$^{-1}$ EGF for 5 min, were subjected to IB. MEFs, mouse embryonic fibroblasts; RT-qPCR, real-time quantitative reverse transcription PCR

*Usp11*$^{-/Y}$ MEFs (Supplementary Fig. 5d). It is also worth noting that the loss of *Usp11* had no significant effect on the subcellular compartmentalization of PTEN (Supplementary Fig. 5e–g).

Given the tumor-suppressive roles of PTEN[3,32], we speculated that the loss of *Usp11* might promote a variety of tumorigenic processes. Indeed, primary *Usp11*$^{-/Y}$ MEFs showed a higher growth rate than their WT counterparts (Fig. 2a, b). The transformation efficiency of adenovirus E1a in combination with the activated Ha-*ras*, SV40 large T antigen (T-Ag) or p16$^{INK4a}$ and p19$^{Arf}$ shRNAs, was also significantly higher in *Usp11*-deficient cells than in WT cultures (Fig. 2c). In vivo mouse allograft generation with these transformed cells confirmed the critical tumor-suppressive function of USP11 (Fig. 2d). When *Usp11*$^{-/Y}$ MEFs were subjected to wound-healing assays, *Usp11*$^{-/Y}$ MEFs were found to close the wound more rapidly than WT cells (Fig. 2e). It is also noted that the increased cell motility upon *Usp11* loss was accompanied by the upregulation of matrix metalloproteinases (MMPs), including gelatinase *Mmp9* and collagenase *Mmp13* (Fig. 2f). Furthermore, *Usp11*-deficient cells showed higher rates of glucose uptake, lactate production, and glutamine consumption than did their WT counterparts (Fig. 2g). In contrast, cells overexpressing USP11 exhibited impaired glucose and glutamine metabolism, accompanied by reduced expression of GLUT1 (glucose transporter type 1) in the plasma membrane fraction (Fig. 2h, i).

**Usp11 knockout enhances tumor progression in TRAMP mice.** The phenotype of *Usp11* knockout mice has not been reported to date. *Usp11*$^-$ mice (*Usp11*$^{-/Y}$ and *Usp11*$^{-/-}$) were viable and born at the expected Mendelian frequency. As decreasing PTEN levels correlate with increased initiation and progression of prostate tumors in the mouse[5,33], we assessed the role of USP11 in prostate tumorigenesis in vivo. Notably, *Usp11*$^{-/Y}$ mice at 11 weeks of age displayed significantly enlarged prostate lobes and increased prostatic epithelial cell proliferation (Fig. 3a–c). We further investigated the consequences of loss of *Usp11* in a mouse model of prostate cancer. To this end, we crossed our *Usp11*$^{-/Y}$ mice with TRAMP (transgenic adenocarcinoma mouse prostate) mice, in order to drive the prostate-specific expression of SV40 T antigen (Ag)[34]. The clinical relevance of the use of SV40 T Ag, which induces oncogenic progression by binding and inactivating *Trp53* and *Rb1* tumor suppressor genes[35], is supported by previous data showing the loss of p53 and Rb in human prostate cancer[36,37]. Moreover, while the TRAMP mouse model is designed to induce the development of prostate tumors, LOH of *Pten* in TRAMP mice led to a significantly increased rate of tumor development, with a subsequent decrease in overall survival[38]. To study the early effects of *Usp11* ablation in the prostate, TRAMP

mice of differing *Usp11* backgrounds were sacrificed at 10 weeks of age and histopathological analysis was performed. TRAMP; *Usp11*$^{-/Y}$ mice showed a significantly higher rate of high-grade prostatic intraepithelial neoplasia (HG-PIN) than age-matched TRAMP;*Usp11*$^{+/Y}$ mice (Fig. 3d, e).

To further analyze the effects of loss of *Usp11* on prostate carcinoma in TRAMP mice, we followed cohorts of TRAMP; *Usp11*$^{+/Y}$ and TRAMP;*Usp11*$^{-/Y}$ mice by magnetic resonance imaging (MRI) analysis. MRI analysis revealed the presence of larger tumor masses in the prostates of 20-week-old TRAMP; *Usp11*$^{-/Y}$ mice than in age-matched TRAMP;*Usp11*$^{+/Y}$ cohorts (Fig. 3f). Histological examination by hematoxylin and eosin staining revealed severe malignant progression in TRAMP; *Usp11*$^{-/Y}$ mice, whereas the prostate glands of age-matched TRAMP;*Usp11*$^{+/Y}$ mice were preserved and the tissue showed a PIN phenotype (Fig. 3g). It is worth noting that TRAMP;*Usp11*$^{-/Y}$ mice exhibited lower levels of PTEN expression and higher phosphorylation levels of AKT than TRAMP control mice (Supplementary Fig. 6). Smooth muscle actin (SMA) staining revealed a highly penetrant invasive prostatic adenocarcinoma in TRAMP;*Usp11*$^{-/Y}$ mice as compared to age-matched TRAMP; *Usp11*$^{+/Y}$ mice (Fig. 3h). Furthermore, multiple tumor nodules metastasized to lymph nodes in TRAMP;*Usp11*$^{-/Y}$ mice (~80% incidence), which was seen at a far lower incidence (< 20%) in TRAMP;*Usp11*$^{+/Y}$ mice (Fig. 3i). These results validate USP11 as a potential X-linked tumor-suppressive factor for prostate cancer initiation, progression, and metastasis.

**USP11-mediated tumor suppression is PTEN dependent.** In prostate cancer, the *ERG* gene is frequently translocated to the *TMPRSS2* promoter region, with the resulting TMPRSS2-ERG fusion protein expressed in ~50% of human prostate cancers[39,40]. As aberrant ERG expression collaborates with *Pten* haplo-insufficiency to promote cancer progression[41,42], we assessed whether *USP11* loss would affect ERG-dependent cellular processes. Depletion of USP11 enhanced growth, invasion, and glucose and glutamine metabolism in ERG-negative 22Rv1 human prostate cancer cells stably overexpressing ERG (22Rv1$^{ERG}$) (Supplementary Fig. 7a–c).

To further study the contribution of USP11 DUB activity to USP11-mediated tumor suppression, we used a human haploid cancer cell line, HAP1, in which the single allele of *USP11* had been knocked out using CRISPR/Cas9 technology. Notably, in HAP1 cells knocked out for *USP11*, reintroduction of WT, but not catalytically inactive CS, USP11 resulted in a decreased cancer cell growth, invasion, and glucose and glutamine metabolism (Fig. 4a–c). Similar results were observed in *Usp11* null MEFs (Supplementary Fig. 7d–f).

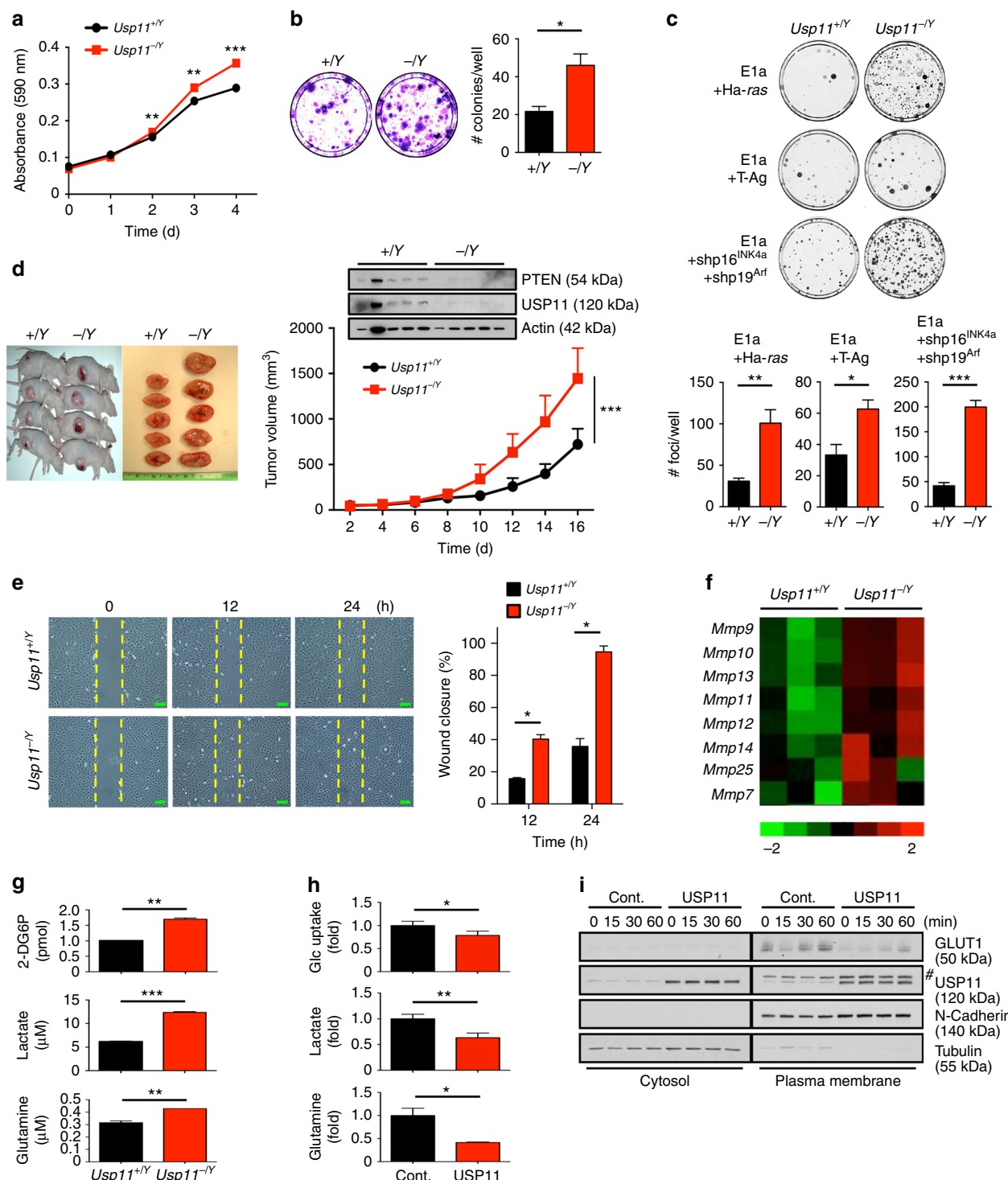

Next, we tested whether USP11 functions as a tumor suppressor through PTEN regulation. Forced expression of USP11 inhibited the growth of PTEN-complemented *PTEN* null PC3 human prostate cancer cells (PC3 *PTEN^WT*)[43], but not PC3 cells expressing the control vector (PC3 *PTEN^-/-*) (Fig. 4d). Overexpression of USP11 also diminished the invasion ability of PC3 *PTEN^WT*, but not PC3 *PTEN^-/-*, cells (Fig. 4e). USP11 overexpression likewise led to a decrease in glucose uptake, lactate production, and glutamine consumption in PC3 *PTEN^WT* cells, while no substantial difference was observed in PC3 *PTEN^-/-* cells (Fig. 4f). Conversely, knocking down USP11 enhanced the cell growth, invasion, and energy metabolism of PC3 *PTEN^WT* cells

but not PC3 *PTEN^-/-* cells (Fig. 4g–i), suggesting that USP11 exerts a PTEN-dependent tumor-suppressive function.

## USP11 is reduced and correlates with PTEN in human cancers.
The tumor-suppressive role of USP11 identified above prompted us to evaluate USP11 expression in cancer patients. Interestingly, a recent transcriptome profiling analysis of human prostate cancer[44,45] revealed that *USP11* transcript expression was downregulated in primary prostate tumors, and its reduction was closely associated with tumor aggressiveness (Fig. 5a). Down-regulation of *USP11* was also observed in human breast tumors[46] (Fig. 5b). We further confirmed the clinical significance of *USP11*

**Fig. 2** Loss of *Usp11* increases cell proliferation, motility, and metabolism. **a** Growth curves of primary wild-type (*Usp11*$^{+/Y}$) and *Usp11*$^{-/Y}$ MEFs. n = 3. **b** Colony-formation efficiency of primary *Usp11*$^{+/Y}$ and *Usp11*$^{-/Y}$ MEFs (left). The number of colonies per well was counted (right). n = 3. **c** Representative plates stained with crystal violet in transformation assays of *Usp11*$^{+/Y}$ and *Usp11*$^{-/Y}$ MEFs with the indicated oncogenes are shown (top). Quantification of the number of transformed foci is also shown (bottom). n = 6. **d** Representative images of allograft tumors of *Usp11*$^{+/Y}$ and *Usp11*$^{-/Y}$ MEFs transformed by E1a + Ha-*ras* oncogenes (top). Lysates from the tumors were subjected to IB (middle). Tumor volumes at different days were measured (bottom). n = 6, p value was determined by ANOVA (***p < 0.001). **e** Representative images of wound-healing assays of *Usp11*$^{+/Y}$ and *Usp11*$^{-/Y}$ MEFs at time points t = 0, 12, and 24 h in the presence of mitomycin C (5 μg ml$^{-1}$) after wound introduction (left). The percentage of wound closure at the indicated time points was determined by the ImageJ 1.46r software (right). Scale bars, 100μm. n = 3. **f** *MMP* family member gene enrichment signature from (**e**) by microarray. n = 3. Heatmap colors represent the relative mRNA expression as indicated in the color key. **g** The rates of glucose uptake, lactate production, and glutamine consumption of *Usp11*$^{+/Y}$ and *Usp11*$^{-/Y}$ MEFs were measured and normalized to cell number. n = 3. **h** The rates of glucose uptake, lactate production, and glutamine consumption of MEFs overexpressing USP11 were measured and normalized to cell number. n = 3. **i** Cytosolic and plasma membrane fractionation of GLUT1 in MEFs overexpressing USP11 treated with insulin (0.5 μg ml$^{-1}$) for the indicated times is shown. # indicates nonspecific band. Error bars represent ± SEM. p Value was determined by Student's t test (*p < 0.05; **p < 0.01; ***p < 0.001). IB, immunoblotting; MEFs, mouse embryonic fibroblasts

downregulation through a cancer patient survival analysis drawn from another available database (http://kmplot.com/analysis/)[31] (Fig. 5c), which suggests that its repression may impact neoplastic malignancies and clinical outcomes.

To determine the clinical relevance of PTEN regulation by USP11, we studied tumor tissue microarrays (TMAs) of human prostate cancer and triple-negative breast cancer (TNBC) samples. First, the fluorescence in situ hybridization (FISH) analysis of *PTEN* DNA status revealed that in our prostate and TNBC TMAs, 85.3% (99 of 116) and 83.5% (86 of 103) had two copies of *PTEN*, respectively (while 13 and 16.5% were hemizygous, 1.7 and 0% exhibited homozygous loss) (Fig. 5d). We then examined whether USP11 expression levels correlated with PTEN protein levels in the TMAs of 99 prostate cancer and 86 TNBC cases presenting two copies of the *PTEN* gene. Notably, an immunohistochemical analysis of human prostate tumors demonstrated a statistically significant positive correlation (r = 0.385, p = 0.012) between USP11 and PTEN protein levels, as 72.9% (35 of 48) of the tumors with low USP11 expression also exhibited low PTEN expression (Fig. 5e). Similarly, a highly positive correlation (r = 0.435, p = 0.008) was observed between USP11 and PTEN protein levels in TNBC samples, as 68.1% (32 of 47) of tumors with low USP11 expression also displayed low PTEN expression. Taken together, these results suggest that in a significant fraction of human malignant tumors, *USP11* is downregulated, and a reduced USP11 expression can function as a mechanism of PTEN inactivation in the absence of *PTEN* genomic loss.

**USP11-mediated cell density-dependent PTEN regulation**. To date, great progress has been made in understanding the cellular functions of PTEN, but the mechanisms of its physiological regulation have not been fully established. Normal cells generally cease proliferation upon reaching confluence, a phenomenon referred to as cell contact inhibition;[47] however, the activation of oncogenes and the inactivation of tumor suppressor genes can prevent contact inhibition[48]. We thus tested the effect of cell density on USP11-mediated PTEN regulation. Intriguingly, PTEN expression was greatly increased in both primary and transformed MEFs grown at a high cell density (Fig. 6a). The observed elevation of PTEN at a high cell density was not due to an increased rate of *Pten* transcription, as Pten mRNA remained the same regardless of cell density (Fig. 6b). Moreover, the PTEN protein turnover rate was higher in sparse cells, and endogenous PTEN immunoprecipitates showed a heavier poly-ubiquitination pattern in cells with a low density than in those with a high density (Fig. 6c, d). We next tested the effect of cell density on USP11-mediated PTEN regulation. Surprisingly, the mRNA levels of USP11, but not of any other known PTEN DUBs, including HAUSP (also known as USP7), USP13, and OTUD3[17,19,20], were far higher in dense cells than in sparse cells (Fig. 6e, f). By

characterizing the outcomes by growing sparse cultures in conditioned media from the dense cultures, we verified that the observed increases in PTEN and USP11 levels were not likely associated with autocrine effects, build-up of metabolites, or changes in pH in culture media (Supplementary Fig. 8a). More importantly, the depletion of USP11 impaired high density-induced PTEN upregulation (Fig. 6g), suggesting that cell density controls the physiological levels of PTEN protein, at least in part, through the transcriptional modulation of *USP11*.

**FOXO acts as a key transcriptional activator of *USP11***. Several lines of evidence have demonstrated that in many genes, density-driven cellular mechanisms induce distinct waves of transcriptional activation[49,50]. To investigate how *USP11* is transcriptionally regulated by cell density, we utilized the TRANSFAC 8.3 program and chromatin immunoprecipitation (ChIP)-sequencing databases available via the UCSC Genome Browser (http://genome.ucsc.edu) to uncover the transcription factors bound on the *USP11* promoter. Among multiple candidates, we were particularly interested in FOXO, because it is known to be a key player in high density-induced gene activation[49] (Fig. 6a and Supplementary Fig. 8b, c). Under high-density conditions two independent Foxo1 shRNAs indeed impaired the elevation of protein and mRNA levels of USP11, but not other DUBs, including HAUSP, USP13, and OTUD3 (Fig. 6h and Supplementary Fig. 8d). Likewise, Foxo1 knockdown significantly decreased the luciferase activity of a *USP11* promoter reporter construct, *pUSP11-Luc*, whereas the overexpression of FOXO1, 3, or 4 strongly promoted the transcriptional activation of *USP11* (Fig. 6i and Supplementary Fig. 8e). It is also noted that mutation of the FOXO-binding site within the *USP11* promoter diminished the effect of FOXO1 (Fig. 6j and Supplementary Fig. 8c). Endogenous FOXO1 protein was bound to a region (~80 bp) encompassing a critical FOXO-binding site within *Usp11* promoter, and the binding of FOXO1 was significantly enhanced by a high cell density (Fig. 6k), suggesting that FOXO proteins could directly activate the transcription of *USP11*. More importantly, ablation of *Foxo1* impaired high density-induced PTEN upregulation in primary MEFs (Fig. 6l), suggesting that FOXO plays an important role in the physiological regulation of PTEN expression.

The activation of FOXO can be controlled by acetylation, and resveratrol (3,4′,5-trihydroxystilbene) induces SIRT1-mediated deacetylation of FOXO[51,52] (Fig. 7a). We, therefore, tested whether resveratrol induces *USP11* transcription through SIRT1/FOXO. Resveratrol increased USP11 protein and mRNA levels, but this effect was diminished by the knockdown of SIRT1 (Fig. 7b and Supplementary Fig. 8f). Knocking down FOXO1 similarly reduced the effects of resveratrol on *USP11* transcriptional activation (Fig. 7c and Supplementary Fig. 8g), suggesting that resveratrol promotes USP11 expression, at least in part, through

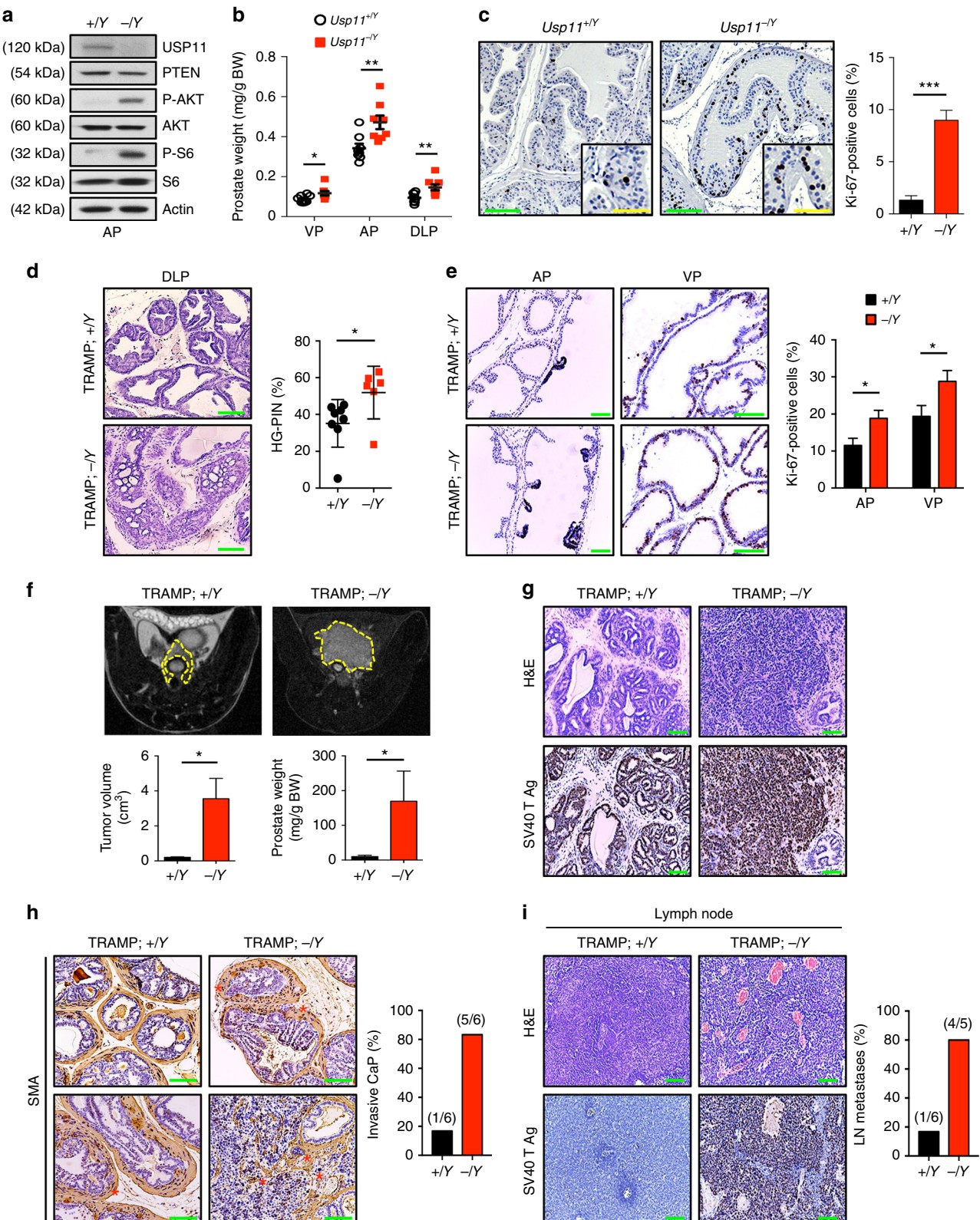

the SIRT1/FOXO pathway. Notably, resveratrol appreciably elevated the PTEN levels, mainly through increasing the stability of PTEN protein (Fig. 7d). It should be noted that resveratrol has been shown to regulate the PTEN/AKT pathway through an androgen receptor (AR)-dependent mechanism[53]. However, our data indicate that the resveratrol-associated elevation of PTEN protein levels is unlikely to be caused by AR-mediated

transcription (Supplementary Fig. 8h, i). More importantly, the elevation of PTEN protein levels by resveratrol was reduced in USP11-depleted cells (Fig. 7e, f), suggesting that resveratrol upregulates PTEN protein expression via the activation of *USP11*.

We also found that the overexpression of active nuclear mutants of FOXO (FOXO1^{T24A,T256A,S319A} or FOXO3^{T32A,S253A,S315A}), which lacked AKT phosphorylation sites[54], led to a marked

**Fig. 3** *Usp11* ablation promotes tumor growth and metastasis in TRAMP mice. **a** Lysates from anterior prostate (AP) of 11-week-old *Usp11*[+/Y] and *Usp11*[-/Y] mice were subjected to immunoblotting. **b** Prostate tissue weight relative to body weight (BW) of 11-week-old *Usp11*[+/Y] ($n = 8$) and *Usp11*[-/Y] ($n = 8$) mice. VP, ventral prostate; AP, anterior prostate; DLP, dorsal-lateral prostate. **c** Ki-67-stained sections of anterior prostate lobes isolated from 11-week-old *Usp11*[+/Y] and *Usp11*[-/Y] mice (left). Quantification of the number of Ki-67-positive cells is shown in right. Green scale bars, 75μm; yellow scale bars, 150μm. **d** H&E-stained sections of dorsal-lateral prostate lobes isolated from 10-week-old TRAMP;*Usp11*[+/Y] ($n = 8$) and TRAMP;*Usp11*[-/Y] ($n = 6$) (left). Quantification of prostatic intraepithelial neoplasia (PIN) is shown in right. Scale bars, 75μm. **e** Ki-67-stained sections of anterior and ventral prostate lobes isolated from 10-week-old TRAMP;*Usp11*[+/Y] ($n = 8$) and TRAMP;*Usp11*[-/Y] ($n = 6$) (left). Quantification of the number of Ki-67-positive cells is shown in the right. Scale bars, 75μm. **f** MRI analysis of 20-week-old TRAMP;*Usp11*[+/Y] ($n = 6$) and TRAMP;*Usp11*[-/Y] ($n = 6$) mice (top), and quantification of prostate tumor volume using an Oxirix Imaging software (bottom, left). Prostate tissue weight of 25-week-old TRAMP;*Usp11*[+/Y] and TRAMP;*Usp11*[-/Y] mice is also shown (bottom, right). **g** Sections of prostate isolated from 25-week-old TRAMP;*Usp11*[+/Y] and TRAMP;*Usp11*[-/Y] mice stained with H&E and anti-SV40 T Ag. Scale bars, 75μm. **h** Sections of prostate isolated from 25-week-old TRAMP;*Usp11*[+/Y] and TRAMP;*Usp11*[-/Y] mice stained with H&E and anti-smooth muscle actin (SMA) (left). Percentage of invasive prostate carcinoma (CaP) in TRAMP;*Usp11*[+/Y] ($n = 6$) and TRAMP;*Usp11*[-/Y] ($n = 6$) mice is shown in the right. Scale bars, 75μm. **i** Sections of lymph nodes isolated from 25-week-old TRAMP;*Usp11*[+/Y] and TRAMP;*Usp11*[-/Y] mice stained with H&E and anti-SV40 T Ag (left). Percentage of lymph node (LN) metastases in TRAMP;*Usp11*[+/Y] ($n = 6$) and TRAMP;*Usp11*[-/Y] ($n = 5$) mice is shown in the right. Scale bars, 75μm. Error bars represent ± SEM. *p* Value was determined by Student's *t* test (\**p* < 0.05; \*\**p* < 0.01; \*\*\**p* < 0.001). H&E, hematoxylin and eosin; MRI, magnetic resonance imaging; TRAMP, transgenic adenocarcinoma mouse prostate

increase in the activity of the *USP11* promoter (Fig. 7g). To further evaluate whether FOXO1 nuclear localization can activate USP11 expression, we treated cells with a specific inhibitor of FOXO1 nuclear export, psammaplysene A (PsA), which is a natural product of the marine sponge *Psammaplysilla* sp[55–58]. Treatment with PsA promoted nuclear retention of FOXO1 as compared to vehicle (Fig. 7h and Supplementary Fig. 8j). Notably, PsA treatment also increased USP11 and PTEN expression levels in a FOXO1-dependent manner (Fig. 7i), suggesting that the enhancement of FOXO nuclear localization leads to substantial activation of USP11 expression.

Next, we sought to determine the clinical relevance of USP11 regulation by FOXO in patients. Importantly, an immunohistochemical analysis of human prostate and TNBC tumors demonstrated a statistically significant positive correlation between FOXO nuclear localization and USP11 expression (Fig. 7j; $p = 0.00064$ and $p = 0.00028$, respectively).

**PTEN auto-regulation by a signaling feedforward mechanism.** Because PI3K/AKT-mediated phosphorylation of FOXO results in its nuclear exclusion and inhibition of transcriptional activity[59], and FOXO activation strongly promotes *USP11* transcription, we speculated that PTEN might be able to regulate itself through a PTEN-PI3K-FOXO-USP11 auto-regulatory loop. We first observed that protein and mRNA levels of USP11, but not any other known PTEN DUBs (such as HAUSP, USP13, and OTUD3), were markedly lower in isogenic *PTEN*-null (*PTEN*[-/-]) HCT116 colon carcinoma cells compared to WT parent (*PTEN*[+/+]) or heterozygous (*PTEN*[+/-]) cells (Fig. 8a, b). A luciferase reporter assay revealed that the loss of *PTEN* similarly inhibited the activity of the *USP11* promoter (Fig. 8c), and notably, the binding of FOXO1 to a region (~80 bp) encompassing a critical FOXO-binding site within *USP11* promoter was impaired in *PTEN*-deficient cells (Fig. 8d). In agreement with these in vitro results, USP11 but not any other known PTEN DUB (including HAUSP, USP13, and OTUD3) has been shown to be downregulated in *Pten* knockout mouse prostates when the incidence of PIN and the increase of cell proliferation are low[43,60] (Fig. 8e), implying a key role for PTEN in the control of USP11 expression in vivo.

Next, to elucidate the contribution of the PTEN/PI3K/AKT signaling pathway to USP11 regulation, we tested USP11 expression upon the blockade of the PI3K/AKT pathway. Notably, the inhibition of PI3K by BKM120 or LY294002 markedly increased USP11 protein and mRNA levels, accompanied by the upregulation of PTEN (Fig. 8f and Supplementary Fig. 9a). Similar observations were made after treatment with the AKT inhibitor MK-2206 (Supplementary Fig. 9b).

Conversely, the treatment with the potent PTEN inhibitor VO-OHpic[4,61,62] led to the reduction of USP11 expression, accompanied by the downregulation of PTEN (Fig. 8g). It is also worth noting that treatment with the proteasome inhibitor MG132 did not result in a further increase in the PTEN protein levels induced by the inhibition of PI3K with BKM120 or ablation of *p85*, the regulatory subunit of PI3K (Fig. 8h, i), which suggests that the PTEN regulatory loop is mediated by the ubiquitin–proteasome pathway. We further examined the effects of the phosphatase activity of PTEN on the regulation of USP11 using *PTEN* null cells complemented with either WT or phosphatase-inactive PTEN (C124S or G129E)[63]. Expression of PTEN[C124S] or PTEN[G129E] phosphatase-inactive mutants led to significant reductions in the levels of protein, mRNA, and promoter activity of *USP11* (Fig. 8j and Supplementary Fig. 9c, d). Taken together, these results suggest that PTEN-PI3K/AKT-FOXO and USP11-PTEN integrate into a PTEN feedforward signaling network (Fig. 8k).

## Discussion
Our findings allow us to reach several relevant conclusions.

First, we have identified the X-linked tumor-suppressor USP11 as an important physiological PTEN DUB that antagonizes the PI3K/AKT signaling pathway. X-linked tumor suppressor genes are of potential significance in tumorigenesis, as LOH or mutation of a single allele can in effect functionally silence a gene[23], and the skewed X inactivation may select toward or against disease[64]. Our *Usp11* knockout strategy supports USP11 as a critical X-linked tumor suppressor that deubiquitinates and stabilizes PTEN. Further, our analysis of cancer patients suggests that reduced *USP11* expression can be a mechanism of PTEN inactivation (or reduced expression) in the absence of *PTEN* deletion. In addition to USP11, HAUSP, USP13, and OTUD3 have all been recently identified as PTEN DUBs. HAUSP is mainly localized in the nucleus and specifically removes the mono-ubiquitination of PTEN to affect its nuclear exclusion[17], while USP13 and OTUD3, which predominantly reside in the cytoplasm or the plasma membrane, reverse the poly-ubiquitination of cytosolic or membrane-bound PTEN protein in a breast cancer-specific context[19,20]. In contrast, here we report that USP11 regulates PTEN poly-ubiquitination and protein stability in both the cytoplasm and nucleus, and in both cancerous (as an important modifier of PTEN levels/activity in prostate cancer) and non-cancerous contexts (as a vector through which cell density controls the physiological dosage of PTEN protein). These findings reveal the latest picture of PTEN DUBs' behaviors (Supplementary Fig. 10), and it will be interesting to further determine their synergism to control PTEN and tumorigenesis in vivo.

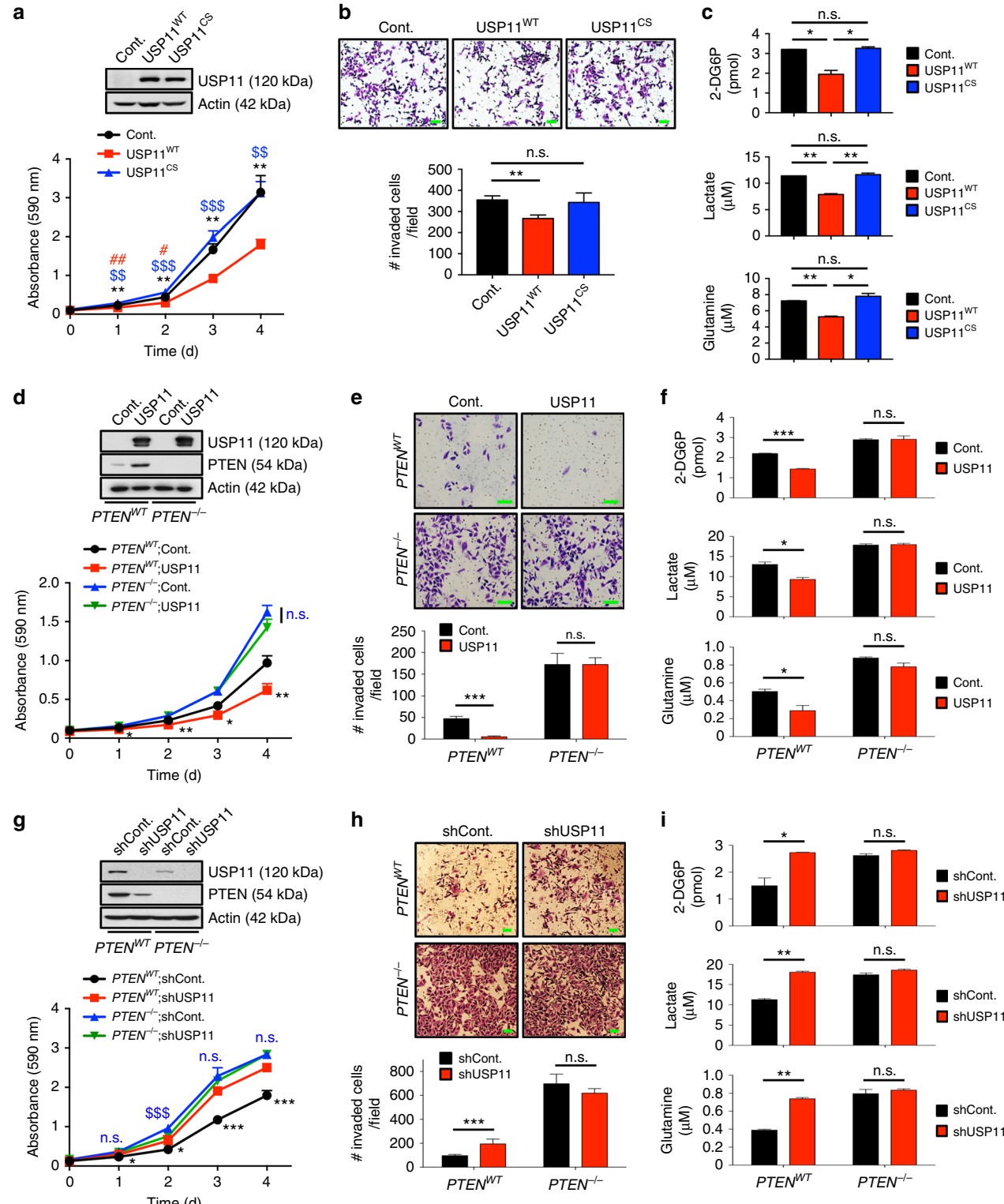

Second, we have found that PTEN auto-regulates itself through the PI3K-FOXO-USP11 feedforward loop to create a PTEN "integrated circuit" that induces tumor suppression. It has been widely reported that important signaling systems are not simple linear pathways, but rather complex dynamic networks in which crosstalks between many players are critical to cellular outcomes. For instance, the p110 delta isoform of PI3K regulates PTEN activity through RhoA-ROCK-dependent signaling[65]. Likewise, activated PI3K has been shown to promote PTEN nuclear exclusion through mTOR signaling[66]. While the precise contribution of the PI3K/AKT pathway to the regulation of PTEN protein expression (and hence tumorigenesis) is yet to be determined, our data provide compelling evidence of a PTEN regulation mechanism whereby a PTEN-PI3K-FOXO-USP11 auto-regulatory loop regulates PTEN protein levels and activity. Indeed, the genetic or pharmacological inhibition of PI3K (through *p85* deletion or BKM120 treatment) increases PTEN protein expression, although these increases are blunted when the

**Fig. 4** USP11 suppresses cancer cell biology in a PTEN-dependent manner. **a** Growth curves of HAP1 cells knocked out (KO) for *USP11* using CRISPR/Cas9 technology expressing wild-type (WT) or catalytically inactive C318S (CS) mutant of USP11. $n = 3$. $**p < 0.01$, $***p < 0.001$, Cont. vs. USP11$^{WT}$; $\$\$p < 0.01$, $\$\$\$p < 0.001$, USP11$^{WT}$ vs. USP11$^{CS}$; $\#p < 0.05$, $\#\#p < 0.01$, Cont. vs. USP11$^{CS}$. **b** Cell-invasion assays of HAP1 cells KO for *USP11* expressing WT or CS mutant of USP11 in the presence of mitomycin C (5 µg ml$^{-1}$) (top). The number of invaded cells per field was quantified (bottom). $n = 3$. **c** The rates of glucose uptake, lactate production, and glutamine consumption of HAP1 cells KO for *USP11* expressing WT or CS mutant of USP11 were measured and normalized to cell number. $n = 3$. **d** Growth curves of PC3 *PTEN*$^{WT}$ and PC3 *PTEN*$^{-/-}$ cells expressing USP11. $n = 3$. $*p < 0.05$, $**p < 0.01$, *PTEN*$^{WT}$; Cont. vs. *PTEN*$^{WT}$; USP11; n.s., non-significant, *PTEN*$^{-/-}$;Cont. vs. *PTEN*$^{-/-}$;USP11. **e** Cell-invasion assays of PC3 *PTEN*$^{WT}$ and PC3 *PTEN*$^{-/-}$ cells ectopically expressing USP11 in the presence of mitomycin C (5 µg ml$^{-1}$) (top). The number of invaded cells per field was quantified (bottom). $n = 3$. **f** The rates of glucose uptake, lactate production, and glutamine consumption of PC3 *PTEN*$^{WT}$ and PC3 *PTEN*$^{-/-}$ cells ectopically expressing USP11 were measured and normalized to cell number. $n = 3$. **g** Growth curves of PC3 *PTEN*$^{WT}$ and PC3 *PTEN*$^{-/-}$ cells expressing USP11 shRNA. $n = 3$. $*p < 0.05$, $***p < 0.001$, *PTEN*$^{WT}$;shCont. vs. *PTEN*$^{WT}$;shUSP11; n. s., non-significant, $\$\$\$p < 0.001$, *PTEN*$^{-/-}$;shCont. vs. *PTEN*$^{-/-}$;shUSP11. **h** Cell-invasion assays of PC3 *PTEN*$^{WT}$ and PC3 *PTEN*$^{-/-}$ cells expressing USP11 shRNA in the presence of mitomycin C (5 µg ml$^{-1}$) (top). The number of invaded cells per field was quantified (bottom). $n = 3$. **i** The rates of glucose uptake, lactate production, and glutamine consumption of PC3 *PTEN*$^{WT}$ and PC3 *PTEN*$^{-/-}$ cells expressing USP11 shRNA were measured and normalized to cell number. $n = 3$. Scale bars, 100 µm. Error bars represent ± SEM. $p$ Value was determined by Student's $t$ test (n.s., non-significant; $*p < 0.05$, $**p < 0.01$, $***p < 0.001$)

proteasome is inhibited. Furthermore, our results reveal that this feedforward mechanism acts as a vector through which cell density controls the physiological dosage of PTEN protein, and, as a result, may enable cells, whether normal or cancerous, to induce PTEN expression during the suppression of cellular proliferation mediated by high density. Because a large fraction of human solid tumors present the loss of only one *PTEN* allele[10,32], we propose that the increased PI3K/AKT activation observed with reduced PTEN levels will lead to the cytoplasmic sequestration of FOXO, reduced USP11 expression, and further PTEN degradation, which in turn sustains PI3K/AKT activation and drives malignant growth in epithelial tissues with heterozygous inactivation of *PTEN*. Thus, our findings provide a rationale for cancer patient stratification based on whether the PTEN-PI3K-FOXO-USP11 feedforward mechanism is 'on' or 'off', which will optimize the personalized therapies by stabilizing PTEN protein in those human cancers that do not exhibit a homozygous (biallelic) loss of *PTEN*.

## Methods

**Mice.** *Usp11* knockout mice were obtained from a knockout mouse cryopreserved sperm (TF2623, Taconic Biosciences) and an oocyte donor. The gene-targeting cassette, containing *lacZ/neo*$^R$, is integrated into exons 3–12 of *Usp11*, generating a dysfunctional allele *Usp11* that is named *Usp11*$^-$. Mice were genotyped by PCR with the following primers: WT, P1: 5′-GTCTGTCTGGGTTCCAGGTTC-3′ and P2: 5′-GCTGACCATGTAGTTCCAAGC-3′; *Usp11*$^-$, P1 and P3: 5′-CCCTAG-GAATGCTCGTCAAGA-3′. Sex determination was performed using the following primers: *Sry*, 5′-CTGTACTCCAAAAACCAGCAAAG-3′; 5′-AGTAAGTAGG-TAAGCTGCTGGTCGT-3′ and *Myog* (Myogenin), 5′-CTGTACTCCAAAAAC-CAGCAAAG-3′; 5′-AGTAAGTAGGTAAGCTGCTGGTCGT-3′. To generate TRAMP;*Usp11*$^{-/Y}$ compound mice, *Usp11*$^{-/Y}$ mice were crossed with TRAMP transgenic mice (Jackson Laboratory). Experimental mice were euthanized at 10 and 25 weeks. At sacrifice, prostatic lobes were removed and weighed. Prostate and lymph node tissues were fixed with 4% paraformaldehyde. Primers for genotyping are TRAMP: 5′-GCGCTGCTGACTTTCTAAACATAAG-3′ and 5′-GAGCT-CACGTTAAGTTTTGATGTGT-3′. All animal experiments in this study were approved by and adhered to the guidelines of the MD Anderson Cancer Center Animal Care and Use Committee.

**Cells.** Primary *Usp11*$^{+/Y}$, *Usp11*$^{-/Y}$, and *Foxo1*$^{fl/fl}$ MEFs were prepared from embryos at 13.5 d of development (E13.5). Early-passage (P1) MEFs were used. 293 T, DU145, PC3, NB4, NIH-3T3, U2OS, MCF-10A, MDA-MB-231, MDA-MB-157, HS578T, BT549, and MDA-MB-468 cell lines were obtained from ATCC (Manassas, VA, USA). HCT116 *PTEN*$^{+/+}$, HCT116 *PTEN*$^{+/-}$ (catalog number: HD 104–057), and HCT116 PTEN$^{-/-}$ (HD 104-004) cell lines and HAP1 cells knocked out for *USP11* using CRISPR/Cas9 technology (HZGH000494c002) were obtained from Horizon Discovery. *Pten*$^{+/+}$ and *Pten*$^{-/-}$ SV40 T-Ag–transformed MEFs are previously described[43]. p85$^{-/-}$ MEFs were kindly provided by L.C. Cantley. All cell lines were independently validated by short tandem repeat DNA fingerprinting and chromosomal analysis by the Characterized Cell Line Facility and the Molecular Cytogenetics Core at the MD Anderson Cancer Center.

**siRNA and shRNA.** All the sources of siRNA duplexes and shRNA constructs used are shown in Supplementary Table 1 and Supplementary Data 1.

**Antibodies.** The antibodies used in the study were anti-PI(3,4,5)P3 (1:200 for immunofluorescence; Z-P345, Echelon Biosciences); anti-PTEN (1:100 for immunofluorescence, 1:200 for immunohistochemistry; ABM-2052, Cascade); anti-USP11 (1:100 for immunofluorescence; KG403, Cosmo Bio); anti-FOXO1 (1:200 for immunofluorescence; 04–1005, Millipore); anti-FOXO1 (1:800 for immunohistochemistry; ab52857, Abcam); anti-FOXO3 (1:800 for immunohistochemistry; 12829, Cell Signaling); anti-FOXO1 (1:50 for ChIP assay; ab39670, Abcam); anti-Ki-67 (1:200 for immunohistochemistry; RM-9106, Lab Vision); anti-SV40 T Ag (1:100 for immunohistochemistry; 554149, BD Pharmingen); anti-SMA (1:200 for immunohistochemistry; ab7817, Abcam); and anti-P-AKT (1:500 for immunohistochemistry; 9271, Cell Signaling). The following antibodies were used for immunoblotting: anti-PTEN (1:1000, 9559), anti-P-AKT (pS473; 1:1000, 9271), anti-P-AKT (pT308; 1:1000, 2965), anti-AKT (pan; 1:1000, 4685), anti-P-FOXO1 (pS256; 1:1000, 9461), anti-FOXO1 (1:1000, 2880), anti-FOXO3a (1:1000, 12829), anti-P-GSK3α (1:1000, 9331), anti-Acetylated lysine (1:1000, 9441), anti-N-Cadherin (1:1000, 4061), and anti-Myc tag (1:5000, 2276) from Cell Signaling; anti-USP11 (1:1000, ab109232), anti-OTUD3 (1:2000, ab107646), anti-GLUT1 (1:1000, ab40084), and anti-Lamin B$_1$ (1:2000, ab16048) from Abcam; anti-USP13 (1:1000, A302-762) and anti-HAUSP (1:5000, A300-033A) from Bethyl Laboratories; anti-Actin (1:10000, A2228), anti-α-Tubulin (1:5000, T8203), anti-Hsp90 (1:1000, H1775), and anti-Flag (1:5000, F1804) from Sigma; anti-GSK3α/β (1:1000, 368662) from Calbiochem; anti-SIRT1 (1:1000, sc-15404) from Santa Cruz; and anti-Poly-ubiquitylated protein (1:1000, PW8805) from Biomol.

**AlphaScreen assay.** Cells were seeded onto 96-well plates at a density of $2 \times 10^3$ cells/well, transfected with 67 mouse DUB siGENOME SMARTpool siRNA library using the Dharmafect siRNA transfection reagent (Supplementary Data 1), starved overnight, and stimulated with 100 nM insulin for 15 m. Cells were collected and cell lysis was prepared according to the manufacturer's instructions. All AlphaScreen assays were performed in triplicate with white ½ Areaplate-96 plates (PerkinElmer). For the detection of phosphorylated AKT proteins, anti-P-AKT-coated AlphaScreen SureFire P-AKT1 (pT308) and P-AKT1/2/3 (pS473) acceptor beads (PerkinElmer) were used. Briefly, 16 µl of cell lysate and acceptor beads in buffer (20 µl) were transferred to each well and incubated for 2 h at room temperature. A total of 8 µl of a 1:50 dilution of the streptavidin donor beads was then added to give a final volume of 44 µl, and the mixture was incubated at room temperature for 2 h. All additions and incubations were made in subdued lighting conditions due to the photosensitivity of the beads, and finally, the assay plates were read with a Synergy Neo multi-mode microplate reader (Biotek).

**Lipid extraction and quantification of PI(3,4,5)P3.** Cells ($1 \times 10^7$) were incubated with cold 0.5 M trichloroacetic acid for 5 m, centrifuged, and resuspended in 5% trichloroacetic acid and 1 mM ethylenediaminetetraacetic acid (EDTA). After removing neutral and acidic lipids, chloroform and 0.1 M HCl were added to the supernatant followed by centrifugation to separate organic and aqueous phases. The organic phase was dried in a vacuum dryer. PI(3,4,5)P3 was quantified using the PI(3,4,5)P3 Mass ELISA kit (Echelon Biosciences). Absorbance was measured at 450 nm using an iMark microplate reader (Bio-Rad).

**In vivo and in vitro deubiquitination assays.** 293T cells were transfected with pCMV-Flag-USP11$^{WT}$ or pCMV-Flag-USP11$^{C318S}$, pcDNA3-HA-Ubiquitin and pRK5-Myc-PTEN and treated with 10 µM MG132 (Sigma) for 4 h. Cells were lysed in 1% sodium dodecyl sulfate (SDS), boiled, and sonicated before diluting in RIPA [25 mM Tris-HCl (pH 7.6), 150 mM NaCl, 1 mM EDTA, 1% NP40, 1% sodium deoxycholate, and 0.1% SDS]. The lysates were immunoprecipitated with anti-Myc tag antibody. The resulting immunoprecipitates were washed five times with RIPA buffer and subjected to immunoblotting with anti-HA (1:1000, MMS-101P; Covance). Alternatively, cells were lysed by incubation for 10 min at 95 °C with two volumes of tris-buffered saline (TBS) containing 2% SDS. After adding eight

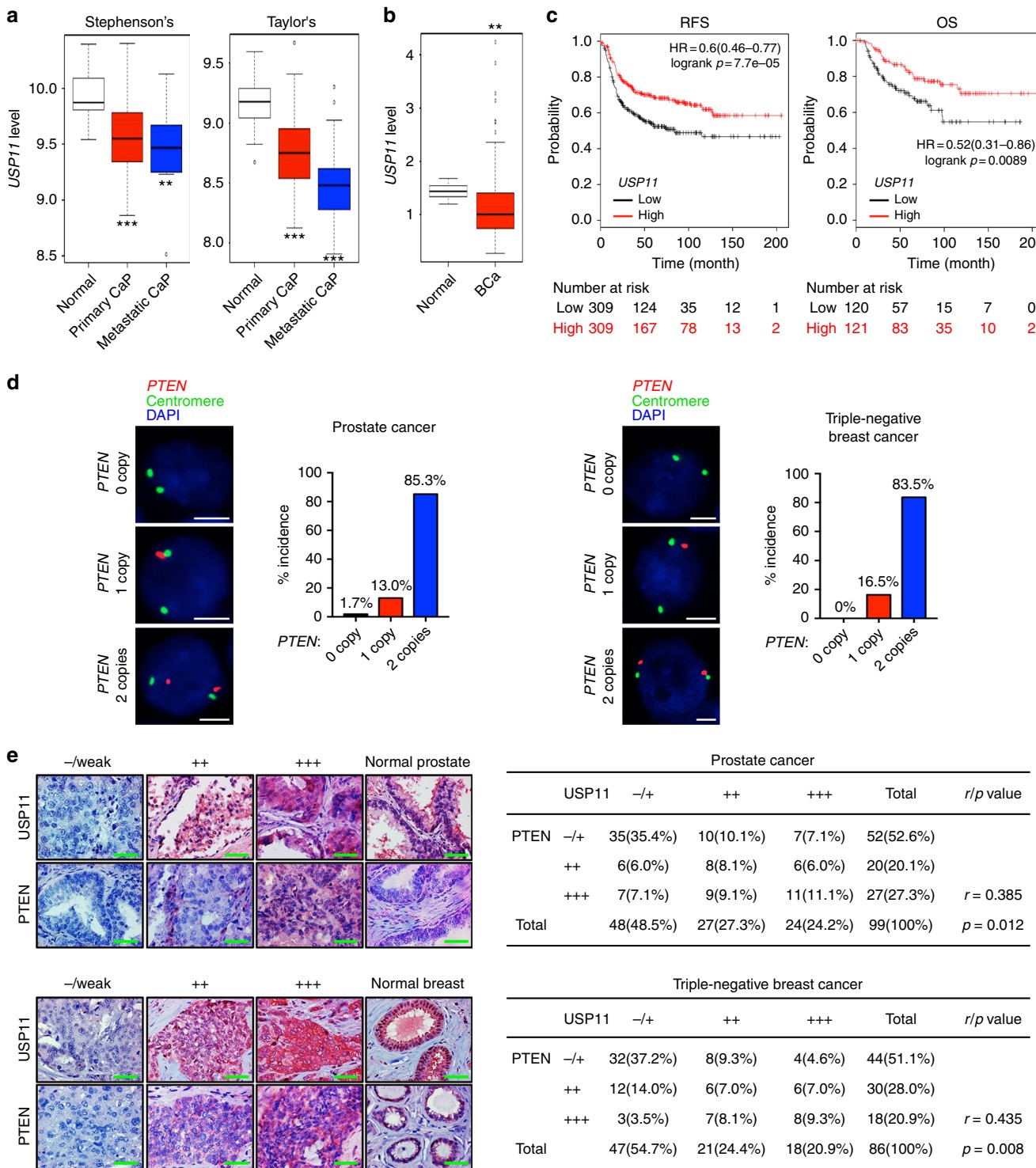

**Fig. 5** USP11 is downregulated and correlates with PTEN in human cancers. **a, b** USP11 expression in human prostate (**a**) and breast (**b**) cancers using a previously published microarray database. Stephenson's CaP, prostate carcinoma (ref.[44], n = 97); Taylor's CaP (GEO: GSE21032, n = 179); Liu's BCa, breast carcinoma (GEO: GSE22820, n = 176). The line in the middle, upper end, and lower end of the boxplot represents the mean, upper, and lower quartile of the relative mRNA level of all samples, respectively. The lines above and below the box are the maximum and minimum values. Data points beyond the whiskers ( >1.5 interquartile ranges) are drawn as individual dots. Error bars represent ± SEM. p Value was determined by Student's t test (**p < 0.01; ***p < 0.001). **c** Online analysis of relapse-free survival (RFS) (n = 618) and overall survival (OS) (n = 241) in human basal-type breast cancer patients with high or low USP11 expression. The number of surviving patients at different time points is indicated below the graphs. p Value was determined by log-rank (Mantel–Cox) test. HR, hazard ratio. **d** Fluorescence in situ hybridization (FISH) analysis of PTEN in human prostate (left, n = 116) and TNBC (right, n = 103) tumor samples. The table indicates the frequency of PTEN immunohistochemical (IHC) status by PTEN FISH status. Correlation between PTEN IHC and PTEN FISH was determined by the PASS Pearson Chi-square test. **e** IHC analysis of USP11 and PTEN in human prostate (top, n = 99) and TNBC (bottom, n = 86) tumor samples (left). Scale bars, 50μm. Correlation between USP11 and PTEN protein levels was determined by the PASS Pearson Chi-square test (right). r, correlation coefficient. TNBC, triple-negative breast cancer

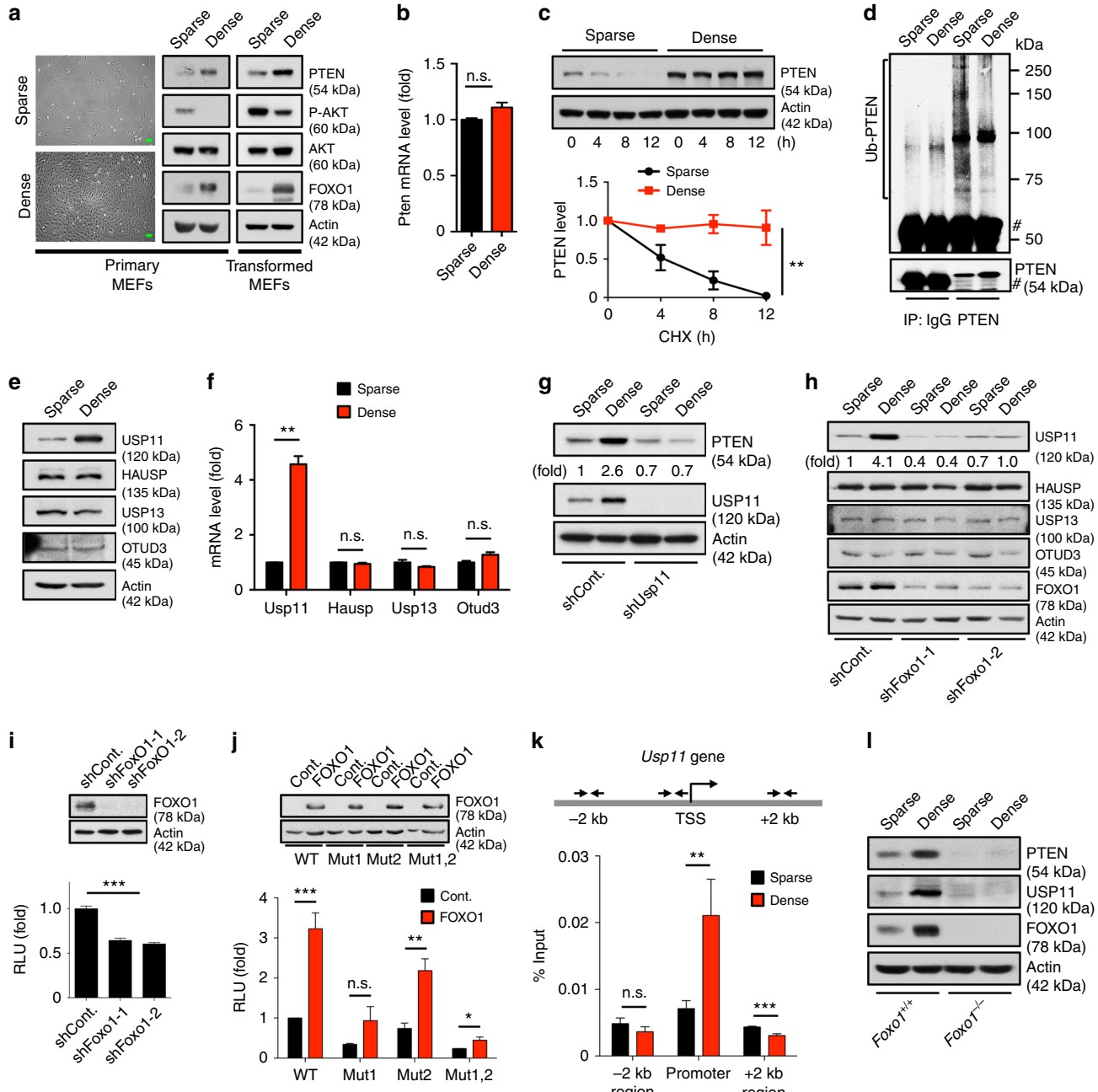

**Fig. 6** USP11-mediated cell density-dependent PTEN regulation. **a** Representative images of sparse- and dense-confluent primary MEFs (left). Immunoblotting (IB) of sparse- and dense-confluent primary (middle) or transformed MEFs (right). Scale bars, 50μm. **b** Total RNAs from (**a**) were subjected to RT-qPCR. $n = 3$. **c** Lysates from sparse- and dense-confluent MEFs treated with cycloheximide (CHX, 100 μg ml$^{-1}$) for the indicated times were subjected to IB (top). PTEN protein levels were quantified by normalizing to the intensity of the actin band (bottom). $n = 3$, $p$ value was determined by ANOVA. **d** Lysates from sparse- and dense-confluent MEFs treated with 10 μM MG132 for 4 h were immunoprecipitated (IP) with anti-PTEN, and the resulting IP were subjected to IB. # indicates the heavy chain of IgG. **e, f** Lysates and total RNAs from sparse- and dense-confluent MEFs were subjected to IB for the indicated proteins (**e**) and RT-qPCR (**f**). $n = 3$. **g** Lysates from sparse- and dense-confluent MEFs expressing Usp11 shRNA were subjected to IB. **h** Lysates from sparse- and dense-confluent MEFs expressing two independent Foxo1 shRNAs were subjected to IB. **i** Luciferase reporter analysis of the *USP11* promoter in NIH-3T3 cells expressing Foxo1 shRNAs. $n = 3$. **j** Luciferase reporter analysis of two potent FOXO-binding site mutants (Mut) of the *USP11* promoter in NIH-3T3 cells expressing FOXO1. $n = 3$. **k** Chromatin levels of FOXO1 at the proximal promoters of mouse *Usp11* were compared between sparse- and dense-confluent MEFs by quantitative ChIP assays. Enrichment of FOXO1 was analyzed with respect to the input control (before IP) and normalized to IgG control. TSS, transcription start site. $n = 3$. **l** Lysates from sparse- and dense-confluent *Foxo1*$^{+/+}$ and *Foxo1*$^{-/-}$ MEFs were subjected to IB. Error bars represent ± SEM. $p$ Value was determined by Student's $t$ test (n.s., non-significant; *$p < 0.05$, **$p < 0.01$, ***$p < 0.001$). ChIP, chromatin immunoprecipitation; MEFs, mouse embryonic fibroblasts; RT-qPCR, quantitative reverse transcription PCR

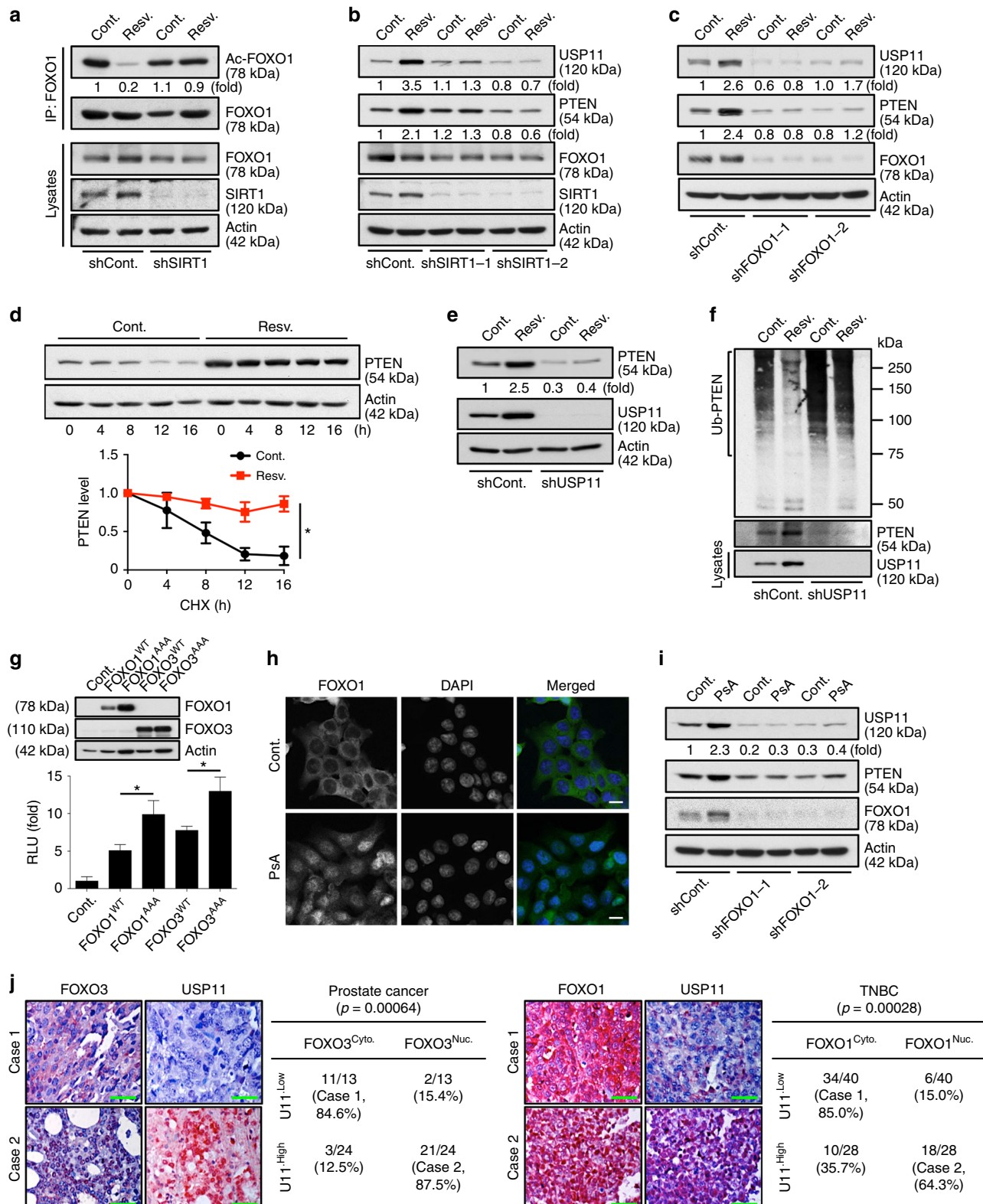

volumes of 1% Triton X-100 in TBS, the lysates were briefly sonicated and then incubated with protein G-agarose beads (Pierce). The beads were removed by centrifugation, and the lysates were immunoprecipitated with anti-PTEN antibody (Cell Signaling). The immunoprecipitates were washed with 0.5 M LiCl in TBS and then subjected to immunoblotting with anti-Ubiquitin (BioMol). For the in vitro deubiquitination assay of PTEN, His-PTEN proteins were incubated with 50 μg ml$^{-1}$ of rabbit E1 (Boston Biochem), 50 μg ml$^{-1}$ of UbcH5 (Boston Biochem), 4 μg ml$^{-1}$ of ubiquitin aldehyde (Calbiochem), along with an ATP regenerating system (7.5 mM creatine phosphate, 1 mM ATP, 1 mM MgCl₂, 0.1 mM

EGTA, rabbit creatine phosphokinase type I (30U ml$^{-1}$ (Sigma)), ubiquitin (10 mg ml$^{-1}$) (Sigma), and 5 μg cell lysate of DU145 to a final reaction volume of 20 μl for 1 h at room temperature. In vitro translated USP11 was then added into the reaction mixture for the indicated times, and the extent of ubiquitination of PTEN was analyzed by sodium dodecyl sulphate-polyacrylamide gel electrophoresis and immunoblotting.

**Real-time quantitative reverse transcription PCR.** Total RNA was isolated with Trizol reagent (Invitrogen) and reverse-transcribed with the PrimeScript reverse

**Fig. 7** FOXO activates the expression of *USP11*, thereby upregulating PTEN. **a** Lysates from DU145 cells expressing SIRT1 shRNA treated with 25 μM resveratrol for 16 h were immunoprecipitated (IP) with anti-FOXO1, and the resulting immunoprecipitates were subjected to immunoblotting (IB). **b, c** Lysates from DU145 cells expressing two independent shRNAs against SIRT1 (**b**) or FOXO1 (**c**) treated with 25 μM resveratrol for 16 h were subjected to IB. **d** Lysates from DU145 cells pre-treated with 25 μM resveratrol for 16 h followed by treatment with cycloheximide (CHX, 100 μg ml$^{-1}$) for the indicated times were subjected to IB. $n = 3$, $p$ value was determined by ANOVA. **e** Lysates from DU145 cells expressing USP11 shRNA treated with 25 μM resveratrol for 16 h were subjected to IB. **f** Lysates from DU145 cells expressing USP11 shRNA and treated with 25 μM resveratrol (Resv.) for 16 h and 10 μM MG132 for the last 4 h before harvesting were immunoprecipitated with anti-PTEN, and the resulting immunoprecipitates were subjected to IB. **g** Luciferase reporter analysis of the *USP11* promoter in NIH-3T3 cells expressing constitutive, active FOXO1$^{T24A,T256A,S319A}$ or FOXO3$^{T32A,S253A,S315A}$. $n = 3$. **h** Immunofluorescence analysis of FOXO1 in *PTEN$^{-/-}$* HCT116 cells treated with 5 μM psammaplysene A (PsA) for 24 h. Scale bars, 10μm. **i** Lysates from *PTEN$^{+/+}$* HCT116 cells treated with 5 μM PsA for 24 h were subjected to IB. **j** Immunohistochemical analysis of FOXO3 or FOXO1 and USP11 in human prostate (left, $n = 37$) and TNBC (right, $n = 68$) tumor samples. Scale bars, 50μm. Correlation between FOXO localization and USP11 protein levels was determined by the PASS Pearson Chi-Square test (right). Error bars represent ± SEM. $p$ Value was determined by Student's $t$ test (*$p < 0.05$, **$p < 0.01$, ***$p < 0.001$). TNBC, triple-negative breast cancer

transcriptase (Takara). Expression of specific mRNAs was determined with a LightCycler (Roche) using the SYBR green PCR master mix (Roche). All the sources of real-time quantitative reverse transcription PCR primers used are listed in Supplementary Table 2.

**Luciferase reporter assay.** Cells were transfected with *pUSP11-Luc* (Genecopeia) or two independent FOXO-binding mutants, *pUSP11-Luc* (Mut1 or Mut2) (Supplementary Table 3) together with a secreted alkaline phosphatase (SEAP) expression vector (Genecopeia) to normalize the transfection efficiency. Cell extracts were processed using the Secrete-Pair Dual Luminescence Assay Kit (Genecopeia) according to the manufacturer's instructions. Luciferase activity was measured with a Synergy Neo multi-mode microplate reader. *Gaussia* luciferase values were normalized to SEAP activity.

**Chromatin immunoprecipitation assays.** ChIP assays were performed using the ChIP kit (Millipore) with the usage of anti-FOXO1 antibody (1:50; ab39670, Abcam) as described[67]. Briefly, cells were fixed with 1% formaldehyde and lysed with lysis buffer (50 mM Tris-HCl (pH 8.0), 10 mM EDTA, 1% SDS, 0.2 mM phenylmethane sulfonyl fluoride, 1 μg ml$^{-1}$ aprotinin, 2.5 μg ml$^{-1}$ leupeptin, 1 μg ml$^{-1}$ pepstatin). The cell lysates were sonicated with a Bioruptor machine (Diagenode) for 12–18 min with cycles of 30 s pulses and 1 min pauses to shear the DNA to a final size of 200–500 bp. After preclearing with protein A beads (Repligen) for 1–2 h, the antibody was added and incubated overnight at 4 °C. Protein A-agarose beads were added and incubated for 1–2 h. The complex was washed with low-salt buffer, high-salt buffer, and LiCl buffer and twice with Tris-EDTA buffer, followed by elution in a buffer containing 1% SDS and 0.1 M NaHCO$_3$. The crosslinks were reversed, and the DNA was purified with a QIA-quick PCR purification kit (Qiagen) and subjected to analysis by real-time qPCR. Chromatin immunoprecipitates for proteins were amplified by qPCR, normalized to input, and calculated as percentage of input. The PCR primers for ChIP assays are listed in Supplementary Table 4.

**Cellular proliferation, migration, and transformation assays.** Typically, cell proliferation, migration, and oncogenic transformation assays were performed as described[8,68,69]. Briefly, growth curves were generated by seeding $2 \times 10^4$ cells into a 12-well plate. Plates were stained with crystal violet at each indicated time point. The dye was extracted with 10% acetic acid followed by plate reading at 595 nm in an iMark Plate Reader (Bio-Rad). For directional cell migration assay, cells were grown to full confluence, serum-starved overnight, and a uniform scratch was made along the center of the plate using a pipet tip. Detached cells were washed out twice, and the remaining cells were then incubated for 12 or 24 h in the presence of mitomycin C (5 μg ml$^{-1}$). Images of cell monolayers were taken at the time indicated under the microscope. The wound wideness was calculated by measuring the mean distance between the margins of the wound in randomly selected fields, directly on photographs. Cell migration was quantified by calculating the area of wound at time points $t = 0$ (time of wound) and $t = 12$ or 24 (12 or 24 h after wound). For the transformation assay, primary *USP11$^{+/Y}$* and *USP11$^{-/Y}$* MEFs (P1) were plated at a density of $6 \times 10^5$ cells into a 10 cm dish and infected with retroviral particles prepared from 293T cells expressing pWZL-E1a (Addgene) with pBabe-Ha-*ras* (Addgene), pMSCV-SV40 T-Ag (Addgene) or pSUPER Retro-p19$^{ARF}$ and –p16$^{INK4a}$ shRNAs (kindly provided by M. Serrano). Foci were scored after 3 weeks by crystal violet staining.

**Glucose uptake, lactate production, and glutamine consumption.** To measure glucose uptake and extracellular lactate, $1–2 \times 10^5$ cells were seeded into a six-well plate. Twelve hours later, the medium was replaced with 3 ml of complete medium and supernatants were collected 24 h later and analyzed for glucose and lactate content using the Glucose colorimetric assay kit II (Bio Vision) and the Lactate colorimetric assay kit II (Bio Vision), respectively, according to the manufacturer's instructions. Glucose consumption was extrapolated by subtracting the measured glucose concentrations in the medium from the original glucose concentration

(25 mM). Both glucose consumption and lactate production are normalized to the total cell number. To measure glutamine consumption, $1–2 \times 10^5$ cells were seeded onto six-well plates. Twelve hours later, the medium was replaced with 3 ml of complete medium and supernatants were collected 24 h later and analyzed for lactate content and glutamine consumption using the Glutamine colorimetric assay kit (Bio Vision), according to the manufacturer's instructions. Glutamine consumption was extrapolated by subtracting the measured glutamine concentrations in the medium from the original glutamine concentration (25 mM). The rates of glucose uptake, lactate production, and glutamine consumption were normalized to cell number.

**Immunofluorescence analysis.** For PI (3,4,5)P3, PTEN, USP11, or FOXO1 immunofluorescence staining, cells were incubated with anti-PI(3,4,5)P3 antibody, anti-PTEN, anti-USP11, or anti-FOXO1. Cells were then washed and incubated with Cy3 anti-mouse secondary antibody (1:5000; Jackson ImmunoResearch) for 1 h. Slides were then mounted in Vectashield containing 4′,6-diamidino-2-phenylindole (Vector Lab). Immunofluorescence images were generated using a Leica DMI 3000B fluorescence microscope (Leica).

**Immunohistochemistry analysis.** Mouse tissue paraffin sections were deparaffinized, rehydrated, and subjected to heat-mediated antigen retrieval in citrate buffer. After blocking the endogenous peroxidase by 3% H$_2$O$_2$, sections were blocked in 10% goat serum and incubated with the following antibodies: anti-PTEN, anti-USP11, anti-P-AKT, anti-Ki-67, anti-SV40 T Ag, and anti-SMA. The signal was detected with biotinylated horseradish peroxidase reagent (Vectastain ABC kit, Vector Lab) and DAB (Vector Lab) according to the manufacturer's instructions. The sections were counter-stained with hematoxylin. The images were acquired with a Leica DM1000 microscope. The stained slides were scanned on Automated Cellular Image System III (Dako, Denmark) for quantification by digital image analysis. The quantification of staining density was performed using an ImagePro software (Media Cybernetics) and calculated based on the average staining intensity and the percentage of positively stained cells.

**Human tumor tissue microarray analysis.** The multiple prostate carcinoma (PR483 and PR803) tumor TMAs were obtained from US Biomax. The study cohort comprised TNBC consecutively ascertained at the MD Anderson Cancer Center (MDACC). All biopsies were evaluated at MDACC, and the histological diagnosis was based on established criteria. Human TMA work was performed in accordance with Institutional Review Board approval at MDACC. TMA slides were incubated with antibodies against PTEN, USP11, FOXO1, or FOXO3 and a biotin-conjugated secondary antibody and then incubated with an avidin–biotin–peroxidase complex. Visualization was performed using 3-amino-9-ethylcarbazole (AEC) chromogen. The TMA cores were scored by two pathologists (W. Xia and Y. Wu) blind to cancer outcomes. According to histologic scoring, the intensity of staining was ranked into one of the four following groups: high (+++), medium (++), low (+), and negative (-).

**Fluorescence in situ hybridization analysis.** FISH assays on TMAs were performed using *PTEN*/Con 10 FISH probes from Empire Genomics. The slides were hybridized with the FISH probes according to the manufacturer's instructions with slight modifications as previously described[70]. Briefly, FISH probes were denatured at 75 °C for 5 min and held at 37 °C for 10–30 min until 10 μl of probe was applied to each sample slide. Slides were coverslipped and hybridized overnight at 37 °C in the ThermoBrite hybridization system (Abbott Molecular). The post-hybridization wash was with 2 × SSC/0.2% Tween-20 at 73 °C for 3 min followed by a brief water rinse. Slides were air dried and then counterstained with Vectashield containing 4′,6-diamidino-2-phenylindole (Vector Lab). The slides were then examined under a fluorescence microscope (Nikon 80i) equipped with multiple filters, and signals were manually counted in 50 cells for each TMA core by two geneticists (A. Multani and S. Pathak) blind to PTEN immunohistochemistry results and cancer outcomes.

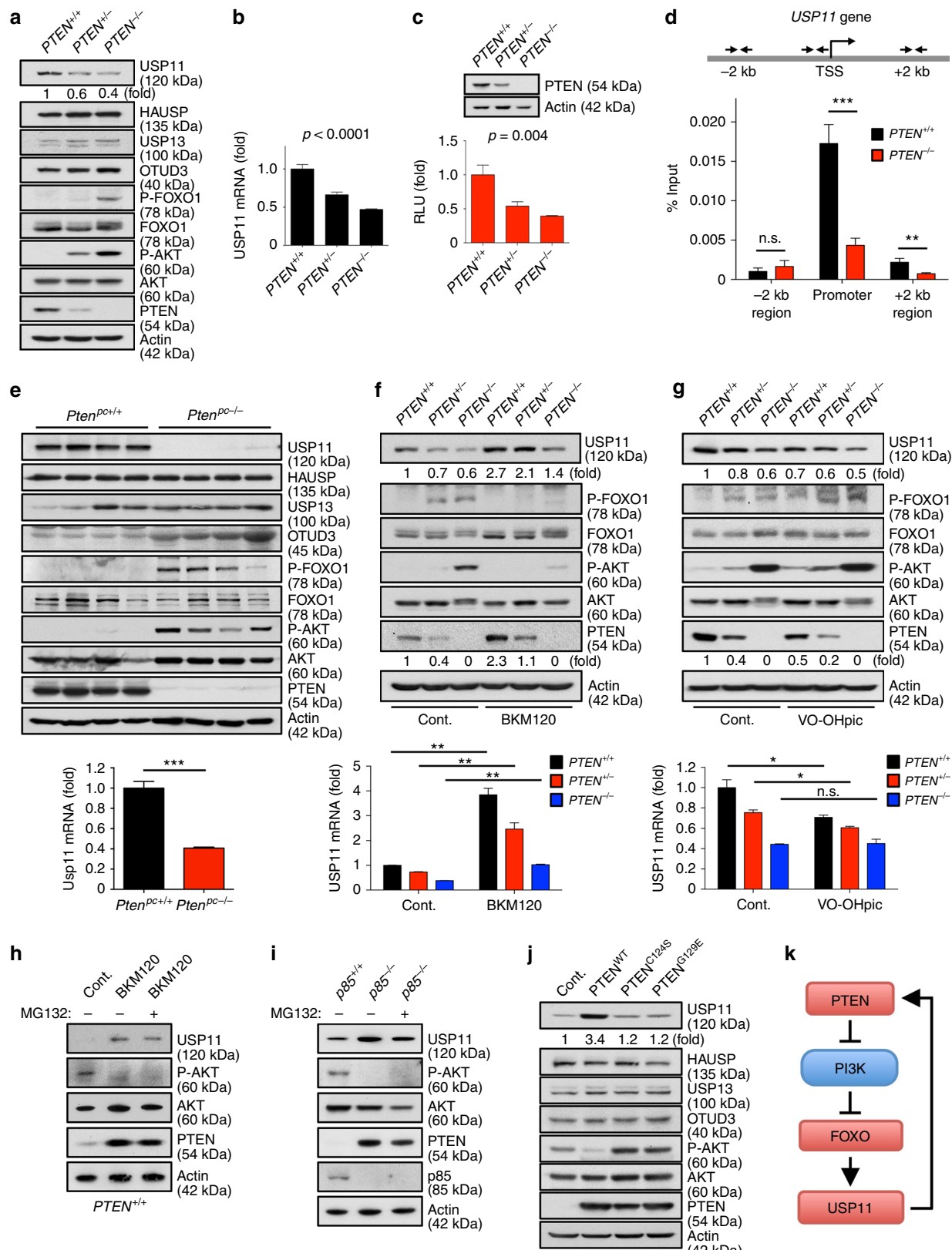

**Cancer biostatistical analysis.** Relative expression levels of *USP11* in prostate or breast cancer patients compared to normal samples were analyzed using previously published microarray database (Stephenson's prostate carcinoma (ref.[44], *n* = 97); Taylor's prostate carcinoma (GEO: GSE21032, *n* = 179); Liu's breast carcinoma (GEO: GSE22820, *n* = 176)). For correlation analysis between *USP11* and prognosis, Kaplan–Meier plotting and log-rank test were performed using a publicly accessible online tool KM Plotter (http://kmplot.com)[31]. Patients were divided into two classes based on *USP11* expression: the high- and low–*USP11* groups were split based on the median value calculated across the entire dataset to generate two groups of equal size. Sample size: RFS, relapse-free survival, *n* = 618; OS, overall survival, *n* = 241.

**Mouse allograft tumor model.** Cells (0.5 × 10^6 cells per site) in suspension were mixed with equal volumes of Matrigel (BD Biosciences) and injected

**Fig. 8** A PTEN-PI3K-FOXO-USP11 auto-regulatory feedforward mechanism. **a, b** Lysates and total RNAs from $PTEN^{+/+}$, $PTEN^{+/-}$, and $PTEN^{-/-}$ HCT116 cells were subjected to immunoblotting (IB) (**a**) and RT-qPCR (**b**). $n = 3$, $p$ value was determined by ANOVA. (**c**) Luciferase reporter analysis of the $USP11$ promoter in $PTEN^{+/+}$, $PTEN^{+/-}$, and $PTEN^{-/-}$ HCT116 cells. $n = 3$, $p$ value was determined by ANOVA. **d** Chromatin levels of FOXO1 at proximal promoters of human $USP11$ were compared between $PTEN^{+/+}$ and $PTEN^{-/-}$ HCT116 cells by quantitative ChIP assays. Enrichment of FOXO1 was analyzed with respect to the input control (before IP) and normalized to IgG control. TSS, transcription start site. $n = 3$. **e** Lysates and total RNAs from anterior prostates of wild-type ($Pten^{pc+/+}$) and $Probasin$-$Cre4;Pten^{loxP/loxP}$ ($Pten^{pc-/-}$) mice at 11 weeks of age ($n = 4$) were subjected to IB (top) and RT-qPCR (bottom). **f** Lysates and total RNAs from $PTEN^{+/+}$, $PTEN^{+/-}$, and $PTEN^{-/-}$ HCT116 cells treated with 500 nM BKM120 for 24 h were subjected to IB (top) and RT-qPCR (bottom). $n = 3$. **g** Lysates and total RNAs from $PTEN^{+/+}$, $PTEN^{+/-}$, and $PTEN^{-/-}$ HCT116 cells treated with 500 nM VO-OHpic for 8 h were subjected to IB (top) and RT-qPCR (bottom). $n = 3$. **h** Lysates from $PTEN^{+/+}$ HCT116 cells treated with 500 nM BKM120 for 24 h together with the absence or presence of MG132 (10 μM, 8 h before harvesting) were subjected to IB. **i** Lysates from $p85^{+/+}$ and $p85^{-/-}$ MEFs treated with 10 μM MG132 for 8 h were subjected to IB. **j** Lysates from $PTEN^{-/-}$ HCT116 cells transfected with wild-type (WT) or phosphatase-inactive $PTEN^{C124S}$ and $PTEN^{G129E}$ subjected to IB. **k** A model for a PTEN-PI3K-FOXO-USP11 auto-regulatory feedforward mechanism. Error bars represent ± SEM. $p$ Value was determined by Student's $t$ test (n.s. non-significant; $^*p < 0.05$, $^{**}p < 0.01$, $^{***}p < 0.001$). ChIP, chromatin immunoprecipitation; IP, immunoprecipitation; MEFs, mouse embryonic fibroblasts; RT-qPCR, quantitative reverse transcription PCR

subcutaneously into 8-week-old male Swiss nu/nu mice (Taconic Biosciences). Measurement of tumor size was performed every other day, and the tumor volume was estimated using the formula: volume = length × width$^2$ × 0.526.

**MRI imaging.** Mice were imaged with abdominal MRI by the MD Anderson Cancer Center Small Animal Imaging Facility. Mice were anesthetized with 2% isoflurane–air mixture. Images were obtained with a 4.7T Brucker MR scanner (40-cm horizontal bore; Brucker Biospin MRI). A double-tuned volume radio-frequency coil was used to scan the abdominal region of the mice. Axial T2-weighted images were acquired using a fast spin-echo sequence with the following parameters: repetition time/effective echo time, 4000/60 ms; echo spacing, 15 ms; number of echoes, 8; field of view, 30 mm × 30 mm; matrix, 128 × 128; slice thickness, 1 mm; slice spacing, 0.25 mm; number of slices, 17 and number of scans, 4 (total scan time was approximately 4 m).

**Statistical analysis.** Data are expressed as the mean ± SEM of the values from the independent experiments performed, as indicated in the corresponding figure legends. The numbers of biological replicates, and what they represent, are indicated in each figure legend. Statistical analysis was performed with SPSS V.20.0 and GraphPad Prism 6. Two-tailed Student's $t$ tests were used for single comparison, and analysis of variance (ANOVA) with Bonferroni post-hoc tests was used for multiple comparisons unless otherwise specified. The correlation coefficients were calculated by the PASS Pearson Chi-Square test. $p$ Values below 0.05 were considered statistically significant.

**Reporting summary.** Further information on the experimental design is available in the Nature Research Reporting Summary linked to this article.

### Data availability
All relevant data are included in the manuscript and available from the authors upon reasonable request. Uncropped gels and blots are available in Supplementary Information (Supplementary Figure 11). A reporting summary for this article is available as a Supplementary Information file.

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

## Acknowledgements

We thank L.C. Cantley and M. Serrano for providing reagents and H.K. Lin, D. Sarbassov, A. Sahin, and C. Logothetis for their advice and critical discussion. We are grateful to T. Garvey for the critical editing of the manuscript. We are thankful to U. Ala for the technical support in cancer biostatistical analysis. We thank J.P. Hwang, C.C. Yeh, and D.H. Agulla for the technical assistance. This work was supported by a Soonchunhyang University Research Fund to S.J. Song and a National Institutes of Health grant (CA196740) to M.S. Song.

## Author contributions

The research was conceived and designed by M.P., S.J.S., and M.S.S. Most experiments were performed by M.P. and Y.Y. W.X., Y.W., and M.-C.H. performed the pathological analysis. J.H.K. and M.G.L. performed ChIP analysis. S.R.S. performed the cancer biostatistical analysis. I.K.V. and H.-H.C. assisted in experiments. E.Z.B. performed MRI analysis. R.G.K. and P.P.P. contributed with critical chemicals and animal models. Data were analyzed by M.P., Y.Y., S.J.S., and M.S.S. The paper was written by M.P., S.J.S., and M.S.S.

## Additional information

**Competing interests:** The authors declare no competing interests.

