## [Peer Review File · Nature Communications]

Editorial Note: Parts of this peer review file have been redacted as indicated to maintain the confidentiality of unpublished data. When text is deleted in rebuttals and referee reports, add “[redacted]” in that location.

Reviewers' comments:

Reviewer #1 (Remarks to the Author):

The authors identify USP11 in a MEF siRNA screen for DUBs that regulate PI3K-Akt signaling and present several lines of evidence that PTEN is a USP11 substrate. This includes co-IP experiments, as well as reduced PTEN stability and increased AKT activation after USP11 knockdown. They then characterize Usp11-deficient mice and show decreased endogenous Pten levels, increased susceptibility to MEF transformation using canonical oncogene combinations and evidence of more aggressive prostate cancer when crossed with TRAMP (SV40 T Aq) mice. The authors then note that confluent cells express higher levels of PTEN and USP11 and propose a feedback pathway of USP11 regulation involving FOXO1. Finally, they include experiments showing increased activity of the PI3K inhibitor BKM120 in combination with resveratrol.

In its current form the manuscript has a number of strengths, particularly implicating USP11 as a regulator of PTEN stability, but there are significant concerns about the later sections that must be addressed.

1. The authors deserve credit for generating and characterizing Usp11-deficient mice. The evidence for enhanced transformation of MEFs by various virus combinations (E1A + RAS, etc) is convincing (Fig 2a-d), but the claims of accelerated prostate cancer progression when crossed to the TRAMP model are not clearly demonstrated (Fig 5). MRI scans alone are not sufficient. We need to see more histological characterization at early and later time points with details on the number of mice analyzed. It is also unclear why the authors selected TRAMP over other models (Tp53, ERG, etc) in which loss of Pten is a rate limiting step for transformation.

2. The correlations between USP11 and PTEN in human tumors are interesting but would be considerably strengthened if accompanied by genomic assessment of PTEN status. This is important since one of the reasons to focus on USP11 is the discrepancy between reduced PTEN expression and PTEN genomic loss in many cancers.

3. The increased levels of PTEN and USP11 expression in confluent cells is intriguing and could reflect a mechanism of physiologic regulation. However, it is unclear why the authors shift to E1A plus SV40 T Aq co-transformation assays to study this question. Furthermore, what is the mechanism by which differences in cell density in culture can confer a stable transformation phenotype that reads out weeks later?? Quite frankly, I find nearly all of the experiments in Fig 6 difficult to explain and a distraction from the main message of the manuscript.

4. The proposed pathway implicating FOXO1 as a PI3K-dependent regulator of USP11 expression, which then feeds back to regulate PTEN stability, is interesting, but the authors need to go further in explaining how this might work. If I understand the model correctly, it functions as a feedforward (not feedback) loop since increased PI3K-AKT activation observed with reduced PTEN levels will lead to cytoplasmic sequestration of FOXO, reduced USP11 expression and further PTEN degradation. If so, the authors need to consider the physiologic rationale for such a feedforward mechanism.

5. The experiments with resveratrol, while potentially interesting, seem to be added at the end of the manuscript to provide some translational implications of the work. In their current form, these experiments are largely descriptive and too preliminary in terms of the characterization of the phenotype. There is much debate about the pharmacological properties of resveratrol (and serious concerns about clinical tolerability from derivatives tested by GSK in patients); therefore, the authors need to perform careful PK/PD measurements and appropriate rescue experiments to prove that the activity observed in the xenograft models is through the proposed FOXO1/USP11 mechanism.

In summary, the manuscript has many positive aspects in terms of the evidence linking USP11 to PTEN stability, but the studies of sparse versus confluent cells and the resveratrol experiments have significant shortcomings.

Reviewer #2 (Remarks to the Author):

This is a novel and interesting finding that places the deubiquitylase, USP11, at the center of a hot area: PTEN and PI3K/AKT and its involvement in cancer.

I will refrain from summarizing the major findings, as the authors know them well.

In principle, I could generally agree with their overall conclusions, IF the data were tabulated properly. As I was going through the data presented in the main figures and in supplemental findings, I noted numerous issues that leave me with much concern about the findings.

I have detailed notes in the attached PDF. If the data are shown as they should be, then the manuscript would merit an even more thorough evaluation on the basis of its interpretations and conclusions. As shown, many of the data are not tabulated or noted properly, and thus I am left unable to come to my own conclusions. Some of the findings are probably biologically relevant, but as presented, I cannot properly review this manuscript since many important data are not shown properly.

Figure 1d: the only conclusion that can be drawn here is that when USP11-CS is over-expressed, Exogenous PTEN is more heavily ub-d. it says nothing about a direct de-ub effect from USP11. Also, this needs to be shown with endogenous PTEN.

1e: what does the asterisk mean, in terms of p-value? does it apply to all points or only the last? are these real biological replicates?

for the p values: this applies to most of the stats, as the authors do not always indicate what they mean, to which data points the asterisks apply, and whether the "N"s are real, independent biological replicates or technical replicates (e.g. same plate transfected multiple times and each well treated as an independent variable, which is not good practice)

figure 2a: these data, just like ALL other cell number data, should be shown in logarithmic scales. I am rather sure that the differences will disappear if that is done, which is the OINLY way to really tabulate cultured cell data. I know it has been standard practice for people to present them this way, but this is fundamentally wrong, and scientifically dubious.

2B: show time course of colony formation, and hopefully blinded numbering. same for 2c.

2D: is there some residual USP11 in tgs? why such variability on the left, non-tg part? loading cannot really seem to account for all of it.

2d, again: unclear how exactly the mouse assays were done.

2D graph: same as in another comment: what do the stats really mean? to what points do the asterisks apply? were these blinded assays?

2E: how were issues from cell division vs. motility addressed?

3A: log scale needs to be used, again.

3D: log scale

4C: blinded assays? easy for someone to squeeze her/his eyes and see differences, or not.

6A: unclear how, exactly, the assays were done

6b: bottom: blinded assays? again, issues with bias from person who counted.

6e: log scale needed

OTUD3 blot in 8A is curious... non-specific band?

9G/H: log scales. 9i: blinded?

suppl 2 c: time course needed for in vitro de-ub assay.

3b:L show a really long exposure of USP11 blot

Reviewer #3 (Remarks to the Author):

In this manuscript, Park et al uncovered a number of significant results that are of great interest to cancer biology and ubiquitin signaling, with important implications for disease control in cancer. Overall, this is a report describing an impressively thorough investigation of posttranslational regulation of the tumor suppressor PTEN by the deubiquitinase (DUB) USP11. Starting with an RNAi screen the authors embarked on further characterization of their hit, USP11's role in PTEN stability and regulation. They find that PTEN, a negative regulator of PI3K/AKT oncogenic pathway, is degraded via polyubiquitination, which is reversed by the action of USP11. But unlike a simple case of protein level control by opposing actions of a ligase and a DUB, they find that PTEN stability is regulated through a feedback loop that involves the FOXO transcription factor. This would be the most remarkable finding of the report, which has a plenty of other interesting data too.

The authors start by validating their hit candidate from the RNAi screen. Three key pieces of evidence indicate that USP11 deubiquitinates PTEN: (1) RNAi suppression of USP11 results in significant loss of PTEN stability (2) USP11 can be pulled down with PTEN, with the interaction involving direct protein-protein interaction between the two proteins and (3) USP11 can remove polyubiquitin chains on PTEN in an in vitro assay. These results identify USP11 as a tumor suppressor by virtue of its stabilizing effect on PTEN. Then the authors unravel a series of studies showing the effect of USP11 on cancer in a number of mice models. USP11 was shown to be downregulated in a number of human cancers (prostate cancer in particular), with the levels of USP11 correlating with that of PTEN in these cancers. An interesting study of the effect of cell density on PTEN-USP11 levels led them to the FOXO transcription factor, an important target of PI3K phosphorylation. Using a reporter assay they show that Foxo1 knockdown decreases luciferase activity of a pUSP11-Luc construct and overexpression of the transcription factor has the opposite effect. These results led to their most interesting finding-- PTEN-PI3K-FOXO-USP11 constitute a regulatory feedback loop to boost stability and tumor suppressor activity of PTEN. In this model of PTEN regulation (autoregulation as the authors refer to somewhere), PTEN activity leads to downregulation of PI3K action, which in turn leads to reduction in FOXO phosphorylation resulting in transcriptional activation of USP11, which by deubiquitinating PTEN completes the feedback loop.

This mode of regulation is quite interesting and I am not sure if there is any other DUB that is part of a similar feedback circuit. Although the cancer related data are quite impressive and important the idea of a DUB in a feedback regulation, particularly through transcriptional control, makes these results the most appealing one to merit publication in a high-profile journal. At the same time, I would have expected the authors to test a key assumption in their model. The regulatory loop assumes a specific role mediated by the ubiquitin-proteasome clearance of PTEN, which unfortunately was not tested. For example, in presence of the proteasome inhibitor (MG132), what would be the effect of a PI3K inhibitor on PTEN protein levels? One would predict that when proteasome is inhibited there would be no need for regulation. I hate to ask more experiments, since there is already a plenty, but I think this type of data might put the regulation loop idea on a more solid footing.

All throughout the paper the authors imply that the DUB activity of USP11 is behind all the effects. I

do not think this has been tested anywhere explicitly. Is the catalytic activity of the DUB required for the regulatory behavior? Most of their studies (after Fig1) deals with USP11 as a protein not as an enzyme, whereas their model assumes a specific enzymatic function. Will it be possible to introduce an inactive DUB mutant, say in USP11 null mutant cell line, to see if the catalytic activity of the DUB is indeed in play?

Responses to the reviewers' comments:

We would like to thank our reviewers for recognizing the importance of our study and for their constructive comments. In response to these suggestions, we have now completed additional experiments that we believe have led to a much-improved manuscript. In the revised version, we believe we have fully addressed the concerns raised during the initial review process, and have added compelling new evidence that strengthens and expands upon our original conclusions.

Specifically, we here provide exciting new data including:

- 1) A more thorough histological characterization with data at early and later time points from the *Usp11*-deficient TRAMP mice analyzed;
- 2) A more comprehensive and detailed analysis of patients vis-à-vis their status regarding reduced PTEN expression vs. *PTEN* genomic loss;
- 3) The critical finding that ablation of *Foxo1* impairs high density-induced PTEN upregulation in primary cells;
- 4) An improved mechanistic explanation for the PI3K-FOXO-USP11-PTEN feed-forward loop, including the findings that: (1) the regulatory loop is mediated by the ubiquitin-proteasome pathway; (2) FOXO is a middleman in the feed-forward loop; and (3) DUB activity by USP11 is essential to the regulatory behavior of the loop.

Additionally, we have now removed from the original manuscript the relevant data regarding the results of resveratrol treatment, which had appeared to legitimize the use of an agent which is now associated with serious concerns regarding its clinical tolerability.

Reviewer #1:

General comments:

In its current form the manuscript has a number of strengths, particularly implicating USP11 as a regulator of PTEN stability, but there are significant concerns about the later sections that must be addressed.

We thank this reviewer for acknowledging the novelty and significance of our study. In this new revised version, we fully address through clarification and experimental demonstration any concerns raised during the initial review process, and in addition have included a plethora of new data to thoroughly support our conclusions.

Specifically, we here provide exciting new data including:

- 1) A more thorough histological characterization with data at early and later time points from the *Usp11*-deficient TRAMP mice analyzed;
- 2) A more comprehensive and detailed analysis of patients vis-à-vis their status regarding reduced PTEN expression vs. *PTEN* genomic loss;
- 3) The critical finding that ablation of *Foxo1* impairs high density-induced PTEN upregulation in primary cells;
- 4) An improved mechanistic explanation for the PI3K-FOXO-USP11-PTEN feed-forward loop, including the findings that: (1) the regulatory loop is mediated by the ubiquitin-proteasome pathway; (2) FOXO is a middleman in the feed-forward loop; and (3) DUB activity by USP11 is essential to the regulatory behavior of the loop.

Additionally, we have now removed from the original manuscript the relevant data regarding the results of resveratrol treatment, which had appeared to legitimize the use of an agent which is now associated with serious concerns regarding its clinical tolerability.

Specific points:

1) The authors deserve credit for generating and characterizing Usp11-deficient mice. The evidence for enhanced transformation of MEFs by various virus combinations (E1A + RAS, etc) is convincing (Fig 2a-d), but the claims of accelerated prostate cancer

progression when crossed to the TRAMP model are not clearly demonstrated (Fig 5). MRI scans alone are not sufficient. We need to see more histological characterization at early and later time points with details on the number of mice analyzed.

We appreciate this reviewer's invaluable suggestion regarding the improvement of our manuscript, and now provide compelling new evidence from a mouse model of prostate cancer demonstrating that USP11 is a critical tumor-suppressive factor for prostate cancer initiation, progression and metastasis (**Figs. 3 & S4**). In our revised manuscript we have included a more thorough histological characterization with data at early and later time points and further details regarding the *Usp11*-deficient TRAMP mice analyzed. First, to study the early effects of *Usp11* ablation, TRAMP mice of differing *Usp11* backgrounds were sacrificed at 10 weeks of age, and histopathological analysis was performed. TRAMP;*Usp11*^{g^Y} mice show a significantly higher rate of high-grade prostatic intraepithelial neoplasia (HG-PIN) than age-matched TRAMP control mice (**Figs. 3d & 3e**). Second, our smooth muscle actin (SMA) staining has revealed highly penetrant invasive prostatic adenocarcinomas in 25-week-old TRAMP;*Usp11*^{g^Y} mice as compared to age-matched TRAMP control mice (**Fig. 3h**). Furthermore, multiple tumor nodules were found to have metastasized to lymph nodes in TRAMP;*Usp11*^{g^Y} mice (~80% incidence), which was seen at a far lower incidence (<20%) in TRAMP control mice (**Fig. 3i**).

It is also unclear why the authors selected TRAMP over other models (Tp53, ERG, etc) in which loss of Pten is a rate limiting step for transformation.

We thank this reviewer for requesting this important clarification. We would also like to respectfully point out that the clinical relevance of the use of SV40 T Ag of TRAMP, which induces oncogenic progression by binding and inactivating *Trp53* and *Rb1* tumor suppressor genes, is supported by previous data showing loss of p53 and Rb in human prostate cancer^{2,3}. Moreover, while the TRAMP mouse model alone is designed to induce the development of prostate tumors, the loss of heterozygosity of *Pten* in TRAMP mice resulted in a significantly increased rate of tumor development, with a subsequent

decrease in overall survival⁴. We now discuss the relevant issues in the newly revised version of our manuscript.

Following the suggestion of this reviewer, we have now assessed the effect of *USP11* loss on ERG-dependent cellular processes in prostate cancer, and now provide new data in our revision showing that depletion of USP11 enhanced growth, invasion, and glucose and glutamine metabolism of a ERG-negative prostate cancer cell line 22Rv1 stably overexpressing ERG (22Rv1^{ERG}) (**Figs. S5a–S5c**).

[Redacted]

2) The correlations between USP11 and PTEN in human tumors are interesting but would be considerably strengthened if accompanied by genomic assessment of PTEN status. This is important since one of the reasons to focus on USP11 is the discrepancy between reduced PTEN expression and PTEN genomic loss in many cancers.

We thank this reviewer for acknowledging the significance of our findings. Importantly, we now provide new evidence clearly demonstrating that low USP11 expression contributes to substantial PTEN reduction without *PTEN* genomic loss in a significant fraction of human cancers, largely through destabilization of PTEN protein. These data include: (1) Fluorescence *in situ* hybridization (FISH) analysis of *PTEN* DNA status in tumor tissues revealing that 85.2% and 83.7% of prostate tumor and TNBC TMAs, respectively, had two copies of *PTEN*, while 13% and 16.3% had hemizygous, and only 1.8% and 0% had homozygous loss in our prostate tumor and TNBC cases, respectively (**Fig. S6**); and (2) After comparing PTEN IHC and FISH results across the prostate tumor and TNBC TMAs, we can now confirm that low USP11 expression contributes to a significant PTEN reduction in tumor cases without *PTEN* genomic loss (**Figs. 5d & 5e**).

3) The increased levels of PTEN and USP11 expression in confluent cells is intriguing and could reflect a mechanism of physiologic regulation. However, it is unclear why the authors shift to E1A plus SV40 T Ag co-transformation assays to study this question. Furthermore, what is the mechanism by which differences in cell density in culture can confer a stable transformation phenotype that reads out weeks later?? Quite frankly, I

find nearly all of the experiments in Fig 6 difficult to explain and a distraction from the main message of the manuscript.

At the suggestion of this reviewer, the results shown in Figs. 6a, b, c, e, f of the original manuscript (including all relevant data regarding the transformation phenotype), along with all discussion of them, have been removed from the revised version. Instead, we have focused on a potential mechanism for physiologic regulation of PTEN dosage, and to this end tested the effect of cell density on USP11-mediated PTEN regulation; we now provide new data showing that PTEN expression is greatly increased in primary cells grown at high cell density (**Fig. 6a**). Moreover, we now provide new evidence that ablation of *Foxo1* impairs high density-induced PTEN upregulation in primary MEFs (**Fig. 6l**), which suggests that FOXO plays an important role in cell-density-dependent PTEN regulation.

4) The proposed pathway implicating FOXO1 as a PI3K-dependent regulator of USP11 expression, which then feeds back to regulate PTEN stability, is interesting, but the authors need to go further in explaining how this might work. If I understand the model correctly, it functions as a feedforward (not feedback) loop since increased PI3K-AKT activation observed with reduced PTEN levels will lead to cytoplasmic sequestration of FOXO, reduced USP11 expression and further PTEN degradation. If so, the authors need to consider the physiologic rationale for such a feedforward mechanism.

We apologize if our original description was unclear. As this reviewer correctly points out, we have now replaced ‘feedback mechanism’ with the new description, ‘**feed-forward** mechanism’ in the new revised version. Furthermore, we now provide an improved mechanistic explanation for the PI3K-FOXO-USP11-PTEN feed-forward loop, including the following insights: (1) treatment with the proteasome inhibitor MG132 does not result in a further increase in the PTEN protein levels induced by PI3K inhibition by BKM120 or ablation of *p85*, the regulatory subunit of PI3K (**Figs. 8h & 8i**), which suggests that the PTEN regulatory loop is mediated by the ubiquitin-proteasome pathway; (2) ablation of *Foxo1* impairs PTEN upregulation (**Fig. 6l**), suggesting that

FOXO acts as a critical component in the PTEN regulatory loop; and (3) in human haploid cancer cells knocked out for *USP11* using CRISPR/Cas9 technology or *Usp11* null MEFs, reintroduction of WT, but not catalytically inactive CS, USP11 resulted in increased PTEN protein levels and decreased cell growth, migration/invasion, and glucose and glutamine metabolism (**Figs. 1e, 4a–4c & S5d–S5f**), indicating that DUB activity by USP11 is essential to the PTEN regulatory loop.

5) The experiments with resveratrol, while potentially interesting, seem to be added at the end of the manuscript to provide some translational implications of the work. In their current form, these experiments are largely descriptive and too preliminary in terms of the characterization of the phenotype. There is much debate about the pharmacological properties of resveratrol (and serious concerns about clinical tolerability from derivatives tested by GSK in patients); therefore, the authors need to perform careful PK/PD measurements and appropriate rescue experiments to prove that the activity observed in the xenograft models is through the proposed FOXO1/USP11 mechanism.

We have taken this criticism to heart, and agree with this reviewer that the resveratrol experiments were overly preliminary and that the concerns recently raised regarding resveratrol could compromise the impact of our study. We have therefore removed from the original manuscript the resveratrol experiments (e.g., Figs. 9g, h, i and S6e, f (all pre-clinically relevant data)), which had appeared to legitimize the use of an agent now associated with serious concerns regarding its clinical tolerability. Instead, in the new revised version we now describe it as a modulator of SIRT1-FOXO signaling for the USP11-PTEN axis (**Figs. 7a–7f**).

[Redacted]

Reviewer #2:

General comments:

This is a novel and interesting finding that places the deubiquitylase, USP11, at the center of a hot area: PTEN and PI3K/AKT and its involvement in cancer. I will refrain from summarizing the major findings, as the authors know them well. In principle, I could generally agree with their overall conclusions, IF the data were tabulated properly. As I was going through the data presented in the main figures and in supplemental findings, I noted numerous issues that leave me with much concern about the findings.

We thank this reviewer for acknowledging the novelty and the significance of our study. In the new revised version, we believe we fully address any and all concerns raised during the earlier review process through clarification and experimental demonstration, and further provide a plethora of new data to thoroughly support our conclusions.

Specifically, we have provided exciting new data including:

- 1) Additional histological characterizations at early and later time points with further details regarding the *Usp11*-deficient TRAMP mice analyzed;
- 2) A more comprehensive and detailed analysis of patients vis-à-vis the status regarding reduced PTEN expression and *PTEN* genomic loss;
- 3) The critical finding that ablation of *Foxo1* impairs high density-induced PTEN upregulation in primary cells;
- 4) An improved mechanistic description of the PI3K-FOXO-USP11-PTEN feed-forward loop, including the new findings that: (1) the regulatory loop is mediated by the ubiquitin-proteasome pathway; (2) FOXO is a middleman in the feed-forward loop; and (3) DUB activity by USP11 is required for the regulatory behavior of the loop.

Specific points:

I have detailed notes in the attached PDF. If the data are shown as they should be, then the manuscript would merit an even more thorough evaluation on the basis of its interpretations and conclusions. As shown, many of the data are not tabulated or noted

properly, and thus I am left unable to come to my own conclusions. Some of the findings are probably biologically relevant, but as presented, I cannot properly review this manuscript since many important data are not shown properly.

We apologize if our descriptions in the original manuscript were unclear. As previously mentioned, in our revised manuscript we fully address, through clarification and experimental demonstration, any and all concerns raised during the earlier review process.

Fig. 1d: the only conclusion that can be drawn here is that when USP11-CS is over-expressed, Exogenous PTEN is more heavily ub-d. it says nothing about a direct de-ub effect from USP11. Also, this needs to be shown with endogenous PTEN.

To fully address this comment, we have provided new data showing that: (1) Wild-type (WT) USP11 deubiquitinates PTEN *in vivo* and *in vitro*, whereas the catalytically inactive C318S (CS) USP11 exerts a diminished ability to deubiquitinate PTEN, and in fact exhibits an even more heavily ubiquitinating effect (**Fig. S2c & S2e**); (2) Likewise, overexpression of WT, but not CS, USP11 increases PTEN protein levels (**Fig. 1e**); and (3) In a human haploid cancer cell line in which the single allele of *USP11* had been knocked out using CRISPR/Cas9 technology, reintroduction of WT, but not CS, USP11 resulted in decreased cancer cell growth, invasion, and glucose and glutamine metabolism (**Figs. 4a–c**). Similar results were observed in *Usp11* null MEFs (**Figs. S5d–f**). These data now clearly demonstrate the contribution of DUB activity by USP11 to USP11-mediated tumor suppression. Additionally, as this reviewer has suggested, we now provide new data showing that WT, but not CS, USP11 reduces endogenous PTEN ubiquitination in cells (**Fig. S2c**).

Fig. 1e: what does the asterisk mean, in terms of p-value? does it apply to all points or only the last? are these real biological replicates?

for the p values: this applies to most of the stats, as the authors do not always indicate what they mean, to which data points the asterisks apply, and whether the "N"s are real, independent biological replicates or technical replicates (e.g. same plate transfected

multiple times and each well treated as an independent variable, which is not good practice).

We apologize if our initial descriptions were unclear. As previously mentioned in the Figure legends and Methods section of the original manuscript, statistical significance was determined using one-way analysis of variance (ANOVA) with Bonferroni post-hoc tests for multiple comparisons of the densitometry data, with p values below 0.05 considered statistically significant. And as previously mentioned in the original Methods section, the data are expressed as the mean +/- SEM of the values from the biological replicates performed, but not technical replicates. The same protocol has been employed in Fig. 2d of the new revised version.

Fig. 2a: these data, just like ALL other cell number data, should be shown in logarithmic scales. I am rather sure that the differences will disappear if that is done, which is the OINLY way to really tabulate cultured cell data. I know it has been standard practice for people to present them this way,m but this is fundamentally wrong, and scientifically dubious.

We would like to respectfully point out that cells in question were first stained with crystal violet solution, the crystal violet dye in each well was then dissolved, and the absorbance was measured by a microplate reader at 590 nm. Nonetheless, as requested, we now show “**Absorbance (590 nm)**” without any transformation of the growth data. The same protocol was employed in Figs. 4a, 4d, 4g, S5a & S5d of the new revised version.

Fig. 2b: show time course of colony formation, and hopefully blinded numbering. same for 2c.

As requested, we now show a time course of colony formation (**Fig. S3f**). We would also like to respectfully point out that signals were manually counted by 3 individuals (M.K.P., H.-H.C. and I.K.V.) who were blind to genotypes and cancer outcomes. The same protocol was employed in Figs. 2c, 4d, 4g, S5a & S5d of the new revised version.

Fig. 2d: is there some residual USP11 in tgs? why such variability on the left, non-tg part? loading cannot really seem to account for all of it.

We thank this reviewer for requesting this clarification. We cannot completely address why there is some residual USP11 protein in the allograft tumor tissues implanted with *Usp11^{gt}* cells. One possible explanation might be an unexpected, although small, level of blood vessel contamination. Indeed, expression of endothelial-specific markers such as *Pecam-1* and *Tie-1* has been found to be far higher in allograft tumor tissues than in cultured MEFs (**Figure 1 for the Reviewer #2**). Nonetheless, we strongly believe that our mouse allograft generation with *Usp11*-deficient transformed cells clearly supports a critical tumor-suppressive function of USP11.

Fig. 2e: how were issues from cell division vs. motility addressed?

As mentioned in the Figure legends of the original manuscript, cells were pre-treated with mitomycin C (5 µg/ml), a potent DNA cross-linker and inhibitor of cell proliferation. This data therefore suggests that cellular proliferation and migration are independently regulated by *Usp11* loss. The same protocol was employed in our cell invasion assays.

Fig. 4c: blinded assays? easy for someone to squeeze her/his eyes and see differences, or not.

We would like to respectfully point out that signals were manually counted by at least 2 pathologists (W.X. and Y.W.) who were blind to genotypes and cancer outcomes. The same protocol was employed in Figs. 3c, 3d, 3e, 3h & 3i of the new revised version.

Fig. 8a: OTUD3 blot in 8A is curious... non-specific band?

We apologize if the OTUD3 WB in the original manuscript was unclear. To fully address any remaining concerns, in the new revised version we now provide new WB data with a brand-new anti-OTUD3 monoclonal antibody (Millipore, MABS1819) in addition to the original polyclonal antibody (Abcam, ab107646).

Fig. S2c: time course needed for in vitro de-ub assay.

We thank this reviewer for requesting this important clarification. Critically, our *in vitro* deubiquitination assays clearly demonstrate that wild-type, but not catalytically inactive C318S, USP11 promotes deubiquitination of PTEN in a time-dependent manner (**Fig. S2e**).

Fig. S3b:L show a really long exposure of USP11 blot.

As requested, in the new revised version we now provide a long exposure of USP11 WB from *Usp11* knockout MEFs.

Figure for Reviewer #2

Figure 1 for the Reviewer #2. Total RNAs from either allograft tumors (n = 5) or cultured *Usp11^{gt/Y}* MEFs (n = 2) were subjected to RT-qPCR. *p* value was determined by Student's *t* test.

Reviewer #3:

General comments:

This mode of regulation is quite interesting and I am not sure if there is any other DUB that is part of a similar feedback circuit. Although the cancer related data are quite impressive and important the idea of a DUB in a feedback regulation, particularly through transcriptional control, makes these results the most appealing one to merit publication in a high-profile journal.

We thank this reviewer for acknowledging the novelty and the significance of our study. In the new revised version, we fully address, through clarification and experimental demonstration, any and all concerns raised during the earlier review process, and in addition provide a plethora of new data to thoroughly support our conclusions.

In particular, we have provided exciting new data including:

- 1) Additional histological characterizations at early and later time points with further details regarding the *Usp11*-deficient TRAMP mice analyzed;
- 2) A more comprehensive and detailed analysis of patients vis-à-vis their status regarding reduced PTEN expression and *PTEN* genomic loss;
- 3) The critical finding that ablation of *Foxo1* impairs high density-induced PTEN upregulation in primary cells;
- 4) An improved mechanistic description of the PI3K-FOXO-USP11-PTEN feed-forward loop, including the new insights that: (1) the regulatory loop is mediated by the ubiquitin-proteasome pathway; (2) FOXO is a middleman in the feed-forward loop; and (3) DUB activity by USP11 is essential to the regulatory behavior of the loop.

Specific points:

1) At the same time, I would have expected the authors to test a key assumption in their model. The regulatory loop assumes a specific role mediated by the ubiquitin-proteasome clearance of PTEN, which unfortunately was not tested. For example, in presence of the proteasome inhibitor (MG132), what would be the effect of a PI3K inhibitor on PTEN protein levels? One would predict that when proteasome is inhibited there would be no

need for regulation. I hate to ask more experiments, since there is already a plenty, but I think this type of data might put the regulation loop idea on a more solid footing.

We appreciate this reviewer's invaluable suggestion regarding the improvement of our manuscript, and we now provide compelling new evidence clearly demonstrating that the PTEN regulatory feed-forward loop is mediated by the ubiquitin-proteasome pathway. These data include: (1) In the presence of the proteasome inhibitor MG132, no further increase was observed in the PTEN protein levels induced by the PI3K inhibitor BKM120 (**Fig. 8h**) and (2) Treatment with MG132 did not result in an increase in the PTEN protein levels induced by ablation of *p85*, the regulatory subunit of PI3K (**Fig. 8i**).

2) All throughout the paper the authors imply that the DUB activity of USP11 is behind all the effects. I do not think this has been tested anywhere explicitly. Is the catalytic activity of the DUB required for the regulatory behavior? Most of their studies (after Fig1) deals with USP11 as a protein not as an enzyme, whereas their model assumes a specific enzymatic function. Will it be possible to introduce an inactive DUB mutant, say in USP11 null mutant cell line, to see if the catalytic activity of the DUB is indeed in play?

We thank the reviewer for requesting this important clarification. To fully address this comment, we now provide new data showing that: (1) Wild-type (WT) USP11 deubiquitinates PTEN *in vivo* and *in vitro*, whereas the catalytically inactive C318S (CS) USP11 exerts a diminished ability to deubiquitinate PTEN, and in fact exhibits an even more heavily ubiquitinating effect (**Fig. S2c & S2e**); (2) Likewise, overexpression of WT, but not CS, USP11 increased PTEN protein levels (**Fig. 1e**); and (3) In a human haploid cancer cell line in which the single allele of *USP11* was knocked out using CRISPR/Cas9 technology, reintroduction of WT, but not CS, USP11 resulted in decreased cancer cell growth, invasion, and glucose and glutamine metabolism (**Figs. 4a–c**). Similar results were observed in *Usp11* null MEFs (**Figs. S5d–f**). These data clearly demonstrate the contribution of DUB activity by USP11 to USP11-mediated tumor suppression.

References

- 1 Colvin, E. K., Weir, C., Ikin, R. J. & Hudson, A. L. SV40 TAg mouse models of cancer. *Semin. Cell Dev. Biol.* **27**, 61-73 (2014).
- 2 Effert, P. J., McCoy, R. H., Walther, P. J. & Liu, E. T. p53 gene alterations in human prostate carcinoma. *J. Urol.* **150**, 257-261 (1993).
- 3 Sarkar, F. H. *et al.* Analysis of retinoblastoma (RB) gene deletion in human prostatic carcinomas. *Prostate* **21**, 145-152 (1992).
- 4 Kwabi-Addo, B. *et al.* Haploinsufficiency of the Pten tumor suppressor gene promotes prostate cancer progression. *Proc. Natl. Acad. Sci. U S A* **98**, 11563-11568, (2001).
- 5 Adams, J. M. *et al.* The c-myc oncogene driven by immunoglobulin enhancers induces lymphoid malignancy in transgenic mice. *Nature* **318**, 533-538 (1985).
- 6 Mori, S. *et al.* Utilization of pathway signatures to reveal distinct types of B lymphoma in the Emicro-myc model and human diffuse large B-cell lymphoma. *Cancer Res.* **68**, 8525-8334 (2008).
- 7 Wendel, H. G. *et al.* Determinants of sensitivity and resistance to rapamycin-chemotherapy drug combinations in vivo. *Cancer Res.* **66**, 7639-7646 (2006).

Reviewers' comments:

Reviewer #1 (Remarks to the Author):

The authors have improved the manuscript significantly but two points still need some clarification.

1) Fig 5 - The authors make the claim that reduced USP11 expression can be a mechanism of PTEN loss (reduced expression) in the absence of PTEN deletion, but the evidence supporting this claim is not clear to me. Panel d shows the correlation between USP11 and PTEN protein expression in human tumors using IHC and panel e shows a correlation between PTEN FISH and PTEN IHC. Isn't the point to show tumors with reduced USP11 and PTEN by IHC, but two intact PTEN alleles by FISH?

2) The evidence for the feedforward loop is convincing but the authors should comment/speculate on the potential physiologic rationale for such a regulatory mechanism. In my mind it seems somewhat counterintuitive since increased PI3K signaling would lead to reduced PTEN levels (due to reduced nuclear FOXO and reduced USP11), which would sustain rather than buffer PI3K activation.

Reviewer #2 (Remarks to the Author):

I appreciate greatly the effort that the authors have put in addressing my concerns. They have done a good job at addressing the technical issues I had raised in the prior round of this manuscript.

I have one major, conceptual issue that I am sure other readers would also have:

As presented, it seems that USP11 is the only important DUB to regulate PTEN via the UPS, although other DUBs have been presented before to do something similar: deubiquitinate PTEN and save it from the proteasome. But, the authors' current data, esp in fig 8, seem to negate the role of other DUBs in the proteasomal-dependent degradation of PTEN. How might the authors square their findings against those published by others in earlier, high-impact journals? At least an in-depth discussion is needed here, if not some additional work to examine the relative role of other DUBs in PTEN in the systems/reagents/techniques employed by the authors. E.g. does USP13 add/subtract to the role of USP11 in saving PTEN from the proteasome, even if it is not through the same "FOXO" pathway? What are the DUBs' relative contributions? These need not be in animals setting, but clarity on the role of these DUBs is, I think, needed at this stage. If this were the first paper published on a DUB's role on PTEN, this would not be required. But this is yet another DUB affecting PTEN. How do all these deubiquitinase stories intersect? When do they?

I am not asking that all DUBs be tested by the authors, but cellular context and how these interactions fit in PTEN is needed at this stage. Evaluating the levels of USP7 etc in PTEN^{-/-} mice does not count towards addressing my point; neither does looking at *usp7* and *usp13* levels in dense/sparse culture. I would probably seek to include USP13 (Nat Cell Bio) OTUD3 (Nat Cell Bio), ataxin3 (Oncogene; albeit not through UPS, still), and/or the USP7/HAUSP network (Nature) in the discussion.

Reviewer #3 (Remarks to the Author):

My concerns have been adequately addressed in this revised version of the manuscript. One of the major shortcomings of the previous version was a lack of a direct connection between the enzymatic

activity of the Usp11 deubiquitinase (Dub) to observed effects on PTEN protein levels and related phenomena. I think the authors have now demonstrated that catalytic activity of the Dub is required for regulation of PTEN stability, thus providing a stronger basis in support of their model. Also, I think that the results of the MG132 experiment presented in Fig 8h and i are consistent with their model. Together, these new results have considerably strengthened their PTEN autoregulation model. It would help to include some discussion of biological significance of the regulatory mechanism proposed by the authors, at least by way of explaining the title of the manuscript. For example, I did not get a good sense of what 'autocorrect' means in the title. One would have to assume that PTEN is a short-lived protein and requires a constant helping hand of Usp11 to boost its levels so that it functions effectively as a tumor suppressor.

Minor points:

1. In Figure 1d, the doublet pattern of the Usp11 band is intriguing. Please provide a clarification of this in the caption. What is HA-Ub here? No explanation is given in the caption regarding the use of HA-Ub.
2. In Figure 2, label g is missing.

Responses to the reviewers' comments:

Reviewer #1:

The authors have improved the manuscript significantly but two points still need some clarification.

We thank the reviewer for recognizing the effort made to address his/her concerns.

1) Fig 5 - The authors make the claim that reduced USP11 expression can be a mechanism of PTEN loss (reduced expression) in the absence of PTEN deletion, but the evidence supporting this claim is not clear to me. Panel d shows the correlation between USP11 and PTEN protein expression in human tumors using IHC and panel e shows a correlation between PTEN FISH and PTEN IHC. Isn't the point to show tumors with reduced USP11 and PTEN by IHC, but two intact PTEN alleles by FISH?

We apologize if our description in the previous version was unclear. To avoid any further confusion, we have now presented PTEN FISH as panel d and USP11 and PTEN IHC as panel e, and have more accurately described the data on pages 11-12 of the revised version.

2) The evidence for the feedforward loop is convincing but the authors should comment/speculate on the potential physiologic rationale for such a regulatory mechanism. In my mind it seems somewhat counterintuitive since increased PI3K signaling would lead to reduced PTEN levels (due to reduced nuclear FOXO and reduced USP11), which would sustain rather than buffer PI3K activation.

We thank this reviewer for acknowledging the significance of our study. We have now included a discussion of the potential physiologic rationale for our PTEN feed-forward mechanism in the Discussion section of the revised version (pages 19-20), as requested.

Reviewer #2:

I appreciate greatly the effort that the authors have put in addressing my concerns. They have done a good job at addressing the technical issues I had raised in the prior round of this manuscript.

We thank the reviewer for recognizing the effort made to address his/her concerns.

How might the authors square their findings against those published by others in earlier, high-impact journals? I would probably seek to include USP13 (Nat Cell Bio) OTUD3 (Nat Cell Bio), ataxin3 (Oncogene; albeit not through UPS, still), and/or the USP7/HAUSP network (Nature) in the discussion.

We apologize if our initial descriptions were unclear. Although we previously mentioned this issue in the Introduction section of the original manuscript, we have now expanded our discussion of the specificities and relevance of USP11 and other previously identified PTEN DUBs to PTEN-dependent tumor suppression in the Discussion section of the revised version (page 18), as requested.

Reviewer #3:

My concerns have been adequately addressed in this revised version of the manuscript.

We thank the reviewer for recognizing the effort made to address his/her concerns.

It would help to include some discussion of biological significance of the regulatory mechanism proposed by the authors, at least by way of explaining the title of the manuscript.

We thank this reviewer for acknowledging the significance of our study. As requested, we have now included a discussion of the biological significance of our PTEN auto-regulatory mechanism in the Discussion section of the revised version (pages 19-20).

Minor points:

1. In Figure 1d, the doublet pattern of the Usp11 band is intriguing. Please provide a clarification of this in the caption. What is HA-Ub here? No explanation is given in the caption regarding the use of HA-Ub.

We thank the reviewer for requesting this important clarification. We would like to respectfully point out that the doublet pattern of the USP11 band shown in Fig. 1d indicates exogenous and endogenous USP11 protein, respectively, and we have made explicit note of this in the Figure legends of the revised version. We apologize that we did not clearly identify HA-Ub in the earlier figure caption. In the Figure legend of the new revised version we now clarify HA-Ub as ‘HA-tagged ubiquitin’.

2. In Figure 2, label g is missing.

We thank the reviewer for requesting this important clarification. We have corrected this omission in the revised version.

Reviewers' comments:

Reviewer #2 (Remarks to the Author):

The authors failed to address my concerns. More critically, they decided to select whatever they felt like addressing by abbreviating my comments to what they sought to address textually.

Below is my prior review in full. I cannot recommend that this manuscript move forward without proper investigation of why the authors decided to abbreviate my response, and without a proper address of my concerns by the authors. What they provided in the Discussion falls short of what I think would be necessary to address this concern, which I include, in full, below:

PRIOR REVIEW COMMENTS

"Reviewer #2 (Remarks to the Author)

I appreciate greatly the effort that the authors have put in addressing my concerns. They have done a good job at addressing the technical issues I had raised in the prior round of this manuscript.

I have one major, conceptual issue that I am sure other readers would also have:

As presented, it seems that USP11 is the only important DUB to regulate PTEN via the UPS, although other DUBs have been presented before to do something similar: deubiquitinate PTEN and save it from the proteasome. But, the authors' current data, esp in fig 8, seem to negate the role of other DUBs in the proteasomal-dependent degradation of PTEN. How might the authors square their findings against those published by others in earlier, high-impact journals? At least an in-depth discussion is needed here, if not some additional work to examine the relative role of other DUBs in PTEN in the systems/reagents/techniques employed by the authors. E.g. does USP13 add/subtract to the role of USP11 in saving PTEN from the proteasome, even if it is not through the same "FOXO" pathway? What are the DUBs' relative contributions? These need not be in animals setting, but clarity on the role of these DUBs is, I think, needed at this stage. If this were the first paper published on a DUB's role on PTEN, this would not be required. But this is yet another DUB affecting PTEN. How do all these deubiquitinase stories intersect? When do they?

I am not asking that all DUBs be tested by the authors, but cellular context and how these interactions fit in PTEN is needed at this stage. Evaluating the levels of USP7 etc in PTEN^{-/-} mice does not count towards addressing my point; neither does looking at usp7 and usp13 levels in dense/sparse culture. I would probably seek to include USP13 (Nat Cell Bio) OTUD3 (Nat Cell Bio), ataxin3 (Oncogene; albeit not through UPS, still), and/or the USP7/HAUSP network (Nature) in the discussion."

Responses to the reviewers' comments:

Reviewer #2:

The authors failed to address my concerns. More critically, they decided to select whatever they felt like addressing by abbreviating my comments to what they sought to address textually. Below is my prior review in full. I cannot recommend that this manuscript move forward without proper investigation of why the authors decided to abbreviate my response, and without a proper address of my concerns by the authors. What they provided in the Discussion falls short of what I think would be necessary to address this concern, which I include, in full, below:

We apologize for the oversight or abbreviation and appreciate the opportunity to further clarify our discovery of the X-linked DUB USP11 as a new and important modifier of PTEN levels/activity among other DUBs. To address this reviewer's concern over purported "biasedness," we have more extensively analyzed the reported sets of DUBs affecting PTEN and now provide numerous concrete and conservatively interpreted data to thoroughly intersect USP11 and other DUBs in PTEN in our systems.

I have one major, conceptual issue that I am sure other readers would also have: As presented, it seems that USP11 is the only important DUB to regulate PTEN via the UPS, although other DUBs have been presented before to do something similar: deubiquitinate PTEN and save it from the proteasome. But, the authors' current data, esp in fig 8, seem to negate the role of other DUBs in the proteasomal-depednent degradation of PTEN. How might the authors square their findings against those published by others in earlier, high-impact journals? At least an in-depth discussion is needed here, if not some additional work to examine the relative role of other DUBs in PTEN in the systems/reagents/techniques employed by the authors. E.g. does USP13 add/subtract to the role of USP11 in saving PTEN from the proteasome, even if it is not through the same "FOXO" pathway? What are the DUBs' relative contributions? These need not be in animals setting, but clarity on the role of these DUBs is, I think, needed at this stage. If

this were the first paper published on a DUB's role on PTEN, this would not be required. But this is yet another DUB affecting PTEN. How do all these deubiquitinase stories intersect? When do they?

We have taken to heart this criticism, and now provide a wide array of compelling and conservatively analyzed data to clearly intersect USP11 and other DUBs affecting PTEN in our systems.

First, we would like to acknowledge that although our screen includes HAUSP, USP13 and ataxin-3, we did not observe significant effects of them on the PI3K/AKT activity and PTEN expression in our initial screen (Fig. S1).

Second, we have examined the interactions among endogenous PTEN, USP11, USP13 and OTUD3 and now provide new data showing that PTEN and DUBs are readily co-immunoprecipitated with each other; however, USP11, USP13 and OTUD3 are not associated with each other (**Fig. S3a**).

Third, we have compared the PTEN-stabilizing ability of USP11, USP13 and OTUD3 by performing side-by-side overexpression and knockdown assays and now include new data showing that USP11, USP13 and OTUD3 exhibit a synergistic effect on PTEN expression (**Figs. S3b, c**).

Fourth, knocking down USP13 or OTUD3 in *PTEN*-proficient human prostate cancer cell line DU145 results in a downregulation of PTEN, consistent with previous reports^{1,2}. Notably, ectopic expression of USP11 in these conditions still upregulates PTEN, suggesting that USP11 could be a more potent regulator of PTEN than other DUBs in our assays. These experiments are now included in **Figs. S3d, e**.

Fifth, our analysis of TCGA datasets from cBioPortal (www.cbioportal.org) and Kmplot (<http://kmpplot.com/analysis/>)³⁻⁶ indicates that the clinical significance of *USP11* is more significant than that of *USP13* or *OTUD3* (**Figure 1 for the Reviewer #2**).

Sixth, in order to understand the contribution of the subcellular distribution of USP11 and other DUBs to PTEN expression, we have performed analysis of PTEN stability in both cytoplasm and nucleus upon depletion of DUBs (**Figs. S3f-i**). This approach shows that USP11, but not USP13 or OTUD3, regulates PTEN expression in the nucleus.

Seventh, we have determined whether USP11 could remove the mono-ubiquitination of PTEN in the nucleus. As we now provide in the **Fig. S3j**, USP11 does not abolish the mono-ubiquitination of PTEN, whereas HAUSP removed the mono-ubiquitination of PTEN (consistent with our previous report⁷), suggesting that USP11 represents a unique DUB that de-polyubiquitinates and stabilizes PTEN protein in the nucleus.

I am not asking that all DUBs be tested by the authors, but cellular context and how these interactions fit in PTEN is needed at this stage. Evaluating the levels of USP7 etc in PTEN^{-/-} mice does not count towards addressing my point; neither does looking at usp7 and usp13 levels in dense/sparse culture. I would probably seek to include USP13 (Nat Cell Bio) OTUD3 (Nat Cell Bio), ataxin3 (Oncogene; albeit not through UPS, still), and/or the USP7/HAUSP network (Nature) in the discussion."

We thank the reviewer for requesting this important clarification. As this reviewer has suggested, we have now provided new data showing the relationships between USP11 and other DUBs affecting PTEN in our systems (**Fig. S3**). We believe that our findings reveal latest picture of PTEN DUBs' behaviors, which is now reflected in the Discussion section of the revised manuscript (**Fig. S9**).

Figure 1 for Reviewer #2

Figure 1 for Reviewer #2. Clinical significance of *USP11*, *USP13* and *OTUD3* in human cancer patients. (a) Genetic alterations of *USP11*, *USP13* and *OTUD3* in human prostate (n = 1013), breast (n = 2051) and lung (n = 1144) cancer patients. (b) Online analysis of relapse-free survival (RFS) in human basal-type breast cancer patients with high or low *USP11* (n = 618), *USP13* (n = 360) or *OTUD3* (n = 618) expression. The number of surviving patients at different time points is indicated below the graphs. *p* value was determined by Log-rank (Mantel-Cox) test. HR, hazard ratio.

References

1. Yuan, L., *et al.* Deubiquitylase OTUD3 regulates PTEN stability and suppresses tumorigenesis. *Nat Cell Biol* **17**, 1169-1181 (2015).

2. Zhang, J., *et al.* Deubiquitylation and stabilization of PTEN by USP13. *Nat Cell Biol* **15**, 1486-1494 (2013).
3. Armenia, J., *et al.* The long tail of oncogenic drivers in prostate cancer. *Nat Genet* **50**, 645-651 (2018).
4. Campbell, J.D., *et al.* Distinct patterns of somatic genome alterations in lung adenocarcinomas and squamous cell carcinomas. *Nat Genet* **48**, 607-616 (2016).
5. Curtis, C., *et al.* The genomic and transcriptomic architecture of 2,000 breast tumours reveals novel subgroups. *Nature* **486**, 346-352 (2012).
6. Gyorffy, B. & Schafer, R. Meta-analysis of gene expression profiles related to relapse-free survival in 1,079 breast cancer patients. *Breast Cancer Res Treat* **118**, 433-441 (2009).
7. Song, M.S., *et al.* The deubiquitylation and localization of PTEN are regulated by a HAUSP-PML network. *Nature* **455**, 813-817 (2008).

REVIEWERS' COMMENTS:

Reviewer #2 (Remarks to the Author):

I appreciate the Authors' responses and the extra work. I would think that the figure prepared specifically for me might be useful as a figure for all to see. I will leave that up to the Authors and the Editor. Otherwise I am satisfied.

Responses to the reviewers' comments (NCOMMS-17-28545C):

Reviewer #2:

I appreciate the Authors' responses and the extra work. I would think that the figure prepared specifically for me might be useful as a figure for all to see. I will leave that up to the Authors and the Editor. Otherwise I am satisfied.

We very much appreciate the reviewer's enthusiasm about the impact of our work. As suggested, we have now included the figure previously provided to this reviewer in Supplementary Figure 4 and cited it in the text of the revised manuscript.